# Flood insurance is a driver of population growth in European floodplains

Max Tesselaar [1] ✉, W. J. Wouter Botzen [1,2], Timothy Tiggeloven [1] & Jeroen C. J. H. Aerts [1,3]

Future flood risk assessments typically focus on changing hazard conditions as a result of climate change, where flood exposure is assumed to remain static or develop according to exogenous scenarios. However, this study presents a method to project future riverine flood risk in Europe by simulating population growth in floodplains, where households' settlement location decisions endogenously depend on environmental and institutional factors, including amenities associated with river proximity, riverine flood risk, and insurance against this risk. Our results show that population growth in European floodplains and, consequently, rising riverine flood risk are considerably higher when the dis-amenity caused by flood risk is offset by insurance. This outcome is particularly evident in countries where flood risk is covered collectively and notably less where premiums reflect the risk of individual households.

It is imperative that climate risk assessments guide climate adaptation policies[1–3]. Traditionally, climate change risk has been assessed with models that mainly focus on changing hazard conditions due to biophysical processes and climate change, exogenous projections of exposed assets and people, and assuming constant vulnerability to project the potential damage of climate hazard[4]. Recent research recognises the dynamic nature of exposure and vulnerability with respect to climate risks, and it emphasises the need to apply these dynamics in models. For example, global flood risk modelling studies have found that socio-economic growth is the dominant driver of increasing riverine flood risk in some regions across the world[5,6]. These large-scale flood risk models consider exposure growth in floodplains based on story lines such as 'shared socio-economic pathways' (SSPs), which are based on several generic scenarios of global developments such as economic growth, political stability, and technological development[7]. Besides exposure, the vulnerability of communities to flooding is also often assumed to remain constant over time[1]. Although these assumptions simplify large-scale (global) climate risk models, they disregard the fact that exposure and vulnerability are intrinsically dynamic[8] and should be modelled as such[9]. Households may, for example, move away from or avoid settling in areas at high risk of flooding[10–12] or apply flood risk reduction measures if they do reside in

these areas[1]. Settling in floodplains may also become more attractive when this area provides aesthetic or recreational amenities[13,14], when flood protection infrastructure is improved[15], or when governments or insurers provide financial compensation after floods[12].

Recent scientific developments address the dynamic interplay between climate hazards, exposure, and vulnerability. Instead of top-down methods, which use static exposure and vulnerability input data to project climate impacts on global or regional scales, bottom-up assessments focus on human behaviour and how individuals dynamically respond in time and space to certain (environmental) conditions[9]. For example, bottom-up approaches are used to project the evacuation behaviour of individuals in the face of disasters[16], analyse adaptation behaviour in response to floods and insurance incentives[17–20], and assess migration flows away from high-risk areas[15,21]. For large-scale climate risk models, it is useful to combine a top-down approach, for example to assess climate hazards, with a bottom-up approach that accounts for behavioural responses to changing hazards.

Large-scale flood risk assessments commonly use generic (top-down) scenarios of population growth, largely because population development is complex and depends on many external factors. However, projecting future flood exposure in these assessments requires a downscaling method where high-level (SSP-)scenarios can

[1]Institute for Environmental Studies, Vrije Universiteit Amsterdam, De Boelelaan 1087, 1081 HV Amsterdam, The Netherlands. [2]Utrecht University School of Economics, Utrecht University, Kriekenpitplein 21-22, 3584 EC Utrecht, The Netherlands. [3]Deltares, Boussinesqweg 1, 2629 HV Delft, The Netherlands. ✉e-mail: max.tesselaar@vu.nl

be applied to forecast exposure on a local scale. Several population projection models[22,23] that are used in high-impact flood risk studies[2,3,5,24,25] assess the suitability of areas to capacitate population growth by identifying geographical features that are likely to encourage or discourage populations to settle, such as elevation, steepness of the terrain, distance from urban centres, and distance from the coast. Importantly, these spatial projections represent a static view of human decision-making in relation to environmental processes, particularly regarding flood risk. A more accurate assessment of developments in flood exposure requires a coupling of human and environmental subsystems[26].

This study develops a bottom-up approach to simulating dynamic processes of flood exposure and vulnerability, as well as the resulting future riverine flood risk, by considering factors that may attract or repel settlement in flood-prone areas. We analyse how population growth in floodplains may differ from existing top-down projections when considering the environmental characteristics of river-floodplains. These include flood risk and amenities associated with river proximity, but also insurance policies, which may reduce the negative impact of flood risk at a floodplain settlement location. Regarding the latter, for instance, the National Flood Insurance Programme in the US has been criticised for encouraging development in flood-prone areas because it offers subsidised coverage against flood damage[27]. In addition, within Europe, the subsidisation of flood insurance premiums in high-risk areas occurs in several countries, including France, Belgium, and Spain, where national flood insurance policies explicitly aim to promote solidarity among households in high- and low-risk areas. On the contrary, several countries, such as Germany and the UK, strive to implement mechanisms that stimulate household-level adaptation, including risk-based premiums[28,29]. This means that policyholders pay a premium that reflects the flood risk of their property. A risk-based insurance premium signals to policyholders the extent to which they are at risk of flooding, which has been found to encourage policyholders to reduce risk[30].

Limiting property development in high-risk areas is perhaps the most effective method to limit population exposure to increasing flood hazard[31]. A variety of policy measures may be implemented to achieve this, including more controlled land-use planning in floodplains[32] or exclusion from flood insurance coverage, as is applied in the UK for buildings constructed after 2009 in high risk areas. The strategy of managed retreat has also been applied more often[33], such as voluntary buyout schemes that governments initiate to facilitate the

relocation of households away from flood-prone areas. Besides such regulatory measures, higher risk-based costs of insurance coverage may also discourage settling in floodplains[12]. This study assesses the extent to which insurance affects exposure development and compares these effects for two types of insurance arrangements that are currently applied in EU countries and the UK (see Methods). The modelling approach developed in this study integrates riverine flood risk, the amenities of river proximity, insurance incentives, and household-level disaster risk-reduction (DRR) in a simulation of household settlement location decisions over time. The simulation applies data or techniques from established models, such as riverine flood risk from a spatially explicit flood risk model[2], and flood insurance premiums and household-level DRR from a partial equilibrium model of European flood insurance markets[19]. Whereas these models assess flood risk and simulate insurance market outcomes, including premiums and incentives for household-level DRR, they do not include an integrated approach to assessing flood exposure growth through location decisions and, in particular, how this may result from insurance policies.

This study shows that population growth in European floodplains and, consequently, rising riverine flood risk are considerably higher when the dis-amenity caused by flood risk is offset by insurance. This outcome is particularly evident in countries where flood risk is covered collectively and notably less where premiums reflect the risk of individual households. With this work, we aim to inform the debate about flood insurance reforms and how insurance markets may contribute to making societies more resilient to climate risks.

## Results

To clearly present the flood exposure projections developed in this study, we consistently compare our results to the data obtained from the 2UP model[22]. This model applies a spatial suitability map to project population growth on a local scale using generic population growth scenarios, such as SSPs. The method developed in this study differs from the 2UP model in that it considers environmental (dis)amenities (including flood risk) and insurance. The population projection using the 2UP model is henceforth referred to as the 'baseline' approach. Figure 1, Panel A shows the projected population growth within floodplains for the period 2010-2050 using the baseline approach. This projection applies population growth according to SSP2, which represents a continuation of historical trends in terms of social, economic, and technological development. In this study, floodplains are

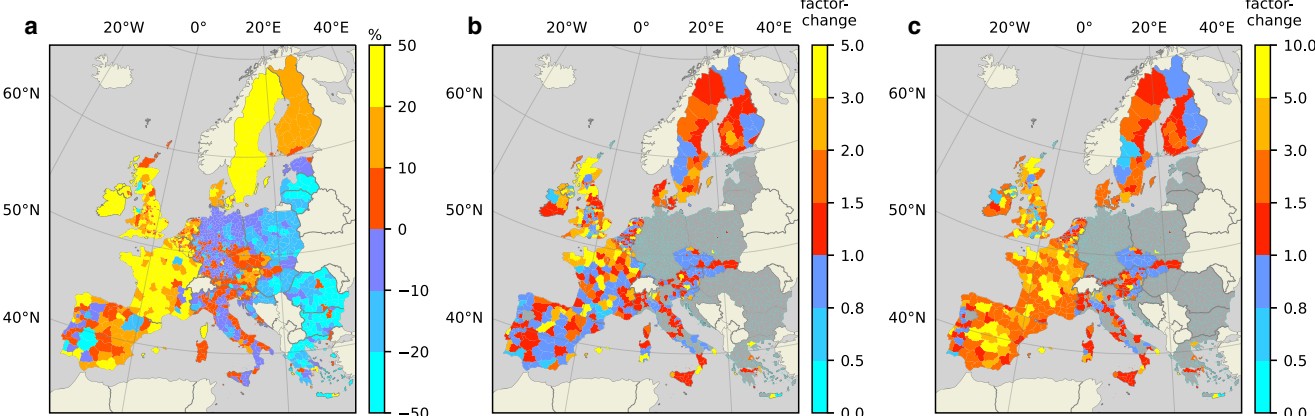

**Fig. 1 | Change in population living in floodplains from 2010 to 2050.**
**a** Projected population development in floodplains from 2010 to 2050 in percentages using the baseline method under SSP2. Blue shades indicate a decline in floodplain populations, while red to yellow shades indicate an increase. **b** The difference from the baseline projection when considering environmental (dis) amenities of floodplains and household-level DRR. **c** The difference with the baseline-projection when considering insurance availability in addition to the determinants included in (**b**). In Panels (**b**, **c**), blue shades indicate a lower flood-plain population compared to the baseline projection, while red to yellow shades indicate a higher population projection. For regions depicted in grey, our method is not applicable due to declining populations across these regions (see Section "Population growth model").

defined as areas that are expected to flood due to river discharge at least once every 100 years, or with a 1% annual probability (see "Methods"). These areas are considered particularly at risk of flooding, and many flood risk management and insurance policy debates specifically concern these areas[18,34]. In this study, flood hazard is defined as the spatial extent and depth of inundation caused by riverine water-levels that occur with a certain probability, and flood risk is defined as the damage caused by such inundation. To enhance visibility, model output throughout this study is aggregated to the NUTS3 level, which is a level of geographic aggregation applied by the EU for specific socio-economic analyses.

Panel B of Fig. 1 shows the extent to which exposure growth projections differ from the baseline when considering environmental amenities associated with rivers and flood risk. Moreover, the analysis for Panel B takes into account the option that households have to reduce flood risk by applying wet- or dry flood-proofing measures (see "Methods"). Flood risk projections used for this assessment apply representative concentration pathway (RCP) 4.5, which represents a future where the 2 °C limit set by the Paris Climate Agreement is met[35, 36]. Projections using alternative SSPs and RCPs are shown in the Supplementary Materials. The results in Panel C follow the same approach, while also considering the potential of insurance to limit the negative impacts of flooding for households. Each panel is discussed in a dedicated section below.

### Floodplain population projections using the baseline-method
Under SSP2, following the baseline approach, Panel A of Fig. 1 shows that population growth in floodplains through 2050 occurs mainly in Western and Northern Europe, while populations are expected to decline considerably in many Eastern European regions. For example, floodplain populations are expected to grow in most regions of France, up to 44%, while only two regions show a projected decline of up to 16%. In several countries, such as Germany, Italy, Austria, and Portugal, floodplain populations are expected to remain more or less stable (over these countries, on average -4%, -0.7%, 5%, and 3%, respectively). Eastern European countries show the most considerable declines in floodplain populations, with the largest declines projected for Bulgaria and Romania (both at approximately -21%). Determinants of population growth under the baseline approach are terrain features such as steepness and roughness, but also proximity to urban centres, which is why the few regions in Eastern Europe where populations do grow are mostly around major cities there (e.g., Bucharest, Warsaw, Athens). A noteworthy observation is that changes in local floodplain populations following the baseline population projection model, as presented in Panel A of Fig. 1, show population projections with patterns similar to those of the larger regions where these floodplains are located (see Methods). This finding is sensible considering that the baseline approach does not specifically assess the suitability for population growth in floodplains. However, there are several areas where floodplain population projections (Panel A of Fig. 1) differ considerably from the overall population projections (see Methods), most notably in the Spanish region of Extremadura. In such cases, projected population growth is largely drawn towards suitable locations, such as urban areas, that are outside the floodplains. In the case of Extremadura, it is likely that attractive settlement locations in the baseline approach are on higher ground, as floodplains make up less than 5% of the landscape (see "Methods").

Contrary to population projections following the baseline approach, recent studies that use satellite imagery show that human settlements worldwide have been growing more rapidly within than outside floodplains in many parts of the world[24,37–39]. In Panels B and C of Fig. 1, we present floodplain population growth projections using the method developed in this study, in which we integrate specific factors that affect the suitability to settle in floodplains.

### Floodplain population growth considering (dis)amenities and household-level DRR
River floodplains are associated with both positive and negative environmental amenities that impact their suitability for human settlement[40,41]. Panel B in Fig. 1 shows how the population projection that considers environmental (dis)amenities in floodplains and household-level DRR deviates from the baseline projection. Positive amenities associated with floodplains are approximated using values derived from studies that apply hedonic pricing techniques, which generally find that positive environmental qualities associated with river proximity decay over distance[42]. The predominant environmental disamenity of river proximity is perceived flood risk, which is simulated in our flood risk model (see Methods). In our simulation, household-level DRR is considered as a strategy for households to reduce flood risk, which is done through a household-level assessment of costs and perceived benefits of wet- and dry flood-proofing measures (see Methods). The option to reduce flood risk may increase the attractiveness of floodplain settlement for some households.

In Panel B of Fig. 1, blue colours indicate lower floodplain populations compared to the baseline approach, while red to yellow colours indicate a higher population projection in our simulation. Whereas a value of 1 implies that our projection is identical to the baseline approach, a value of 2 indicates that our projection is twice as high. The median value of data plotted in Panel B equals 1.1, which implies that in most regions floodplain populations are higher than the baseline projection. However, as blue is also a prominent colour in Panel B, for many regions, the baseline approach does seem to overestimate floodplain population growth when compared to our projections, which means that the flood risk outweighs the amenities of floodplain settlement in these regions.

In countries such as France, Spain, and the UK, there are stark contrasts between regions regarding the projected deviations from the baseline. A driver of these differences is flood risk. In Spain, for example, where three regions are projected to have considerably higher population growth in floodplains compared to the baseline, the annual flood risk per household for each region in 2050 is less than €30 (about €50,000 for each region as a whole), which is a fraction compared to the average per household located within floodplains for the whole of Spain (approximately €400). The same effect can be found in the UK, where, for example, higher population growth and modest annual flood risk per household intersect in several regions of Wales and Northern Scotland. Evidence of the opposite effect, meaning higher flood risk leading to lower population growth in floodplains, is also apparent, most notably for regions in Ireland and central Sweden. In central Sweden, shown in blue, where population growth is projected to be lower compared to the baseline, the annual flood risk per household in 2050 is approximately €700, which is substantially higher than the Swedish average of €270.

In our model, flood risk does not always have a considerable impact on population growth in floodplains. In southern Spain, for example, the flood risk that households face is relatively high, while population projections do not differ noticeably from the baseline projection. The main reason for this result is that general population growth projections for these regions is low, meaning that the potential change from the baseline projection is lower compared to regions where population growth projections are high.

### Floodplain population growth considering insurance coverage
It is rare for households experiencing flood damage to have to completely finance the reconstruction themselves. Flood insurance is available in all EU countries and the UK, which can cover the potential damages against a premium. Panel C in Fig. 1 shows deviations from the baseline projection when households have the option to insure against flood risk.

An important element of this analysis is the flood insurance premium. Although insurers set premiums to cover the risk of flooding over time, often, premiums do not accurately reflect the risk of an individual. Instead, premiums can cover potential losses over a larger geographical space. This means that flood risk in high-risk areas may be partially subsidised by households residing in areas with lower risk. Therefore, if premiums are not risk-reflective, as they are in countries such as France, Belgium, and Spain, the benefits of residing in a river floodplain are more likely to outweigh the costs. This is why positive deviations from the baseline are generally seen in these countries in Panel C of Fig. 1. In the remaining countries, there are more regions where our population projections are lower than the baseline results. In these countries, flood insurance premiums are, to varying degrees, risk-reflective, which means that the costs related to flooding for floodplain residents with insurance coverage approaches the actual projected flood damage. The main difference is that premiums spread flood risk over time into an annual cost, while the damage associated with a flood is likely much larger, although less frequent. Risk-averse individuals will seek a mean-preserving spread of losses, meaning that such people likely prefer a lower certain loss over a larger uncertain loss with the same expected value. This implies that a household facing the choice to settle on a floodplain or on higher ground will most likely choose the floodplain when premiums are insensitive to risk, less likely when premiums are risk-reflective, and least likely without insurance availability.

The assessment done for Panel C in Fig. 1 assigns each country one of the insurance premium types described above based on their actual flood insurance market arrangements, as reviewed in[20] and presented in "Methods".

Panel C of Fig. 1 shows that population growth in floodplains is generally higher than in Panel B. As expected, insurance availability makes the floodplain a more attractive location to settle, causing higher population growth in floodplains compared to both the baseline and the scenario without insurance availability. Over all regions included in the analysis, the mean deviation from the baseline is 2.5, which means that floodplain population growth may be more than twice as high as baseline predictions when considering flood insurance availability, in addition to elements introduced in Panel B. However, this average value is strongly impacted by certain regions where population growth through 2050 is projected to be up to 10 times higher in Panel C. Interestingly, some of these are regions where projections without insurance availability (Panel B) are lower than the baseline (e.g., Deux-Sèvres in France, Cuenca in Spain, West-Surrey in the UK), which indicates that insurance can have a considerable impact on households' settlement location decisions.

Moreover, it is clear that insurance policy design matters. Our population growth projections in floodplains are consistently and substantially higher in France and Belgium, where premiums are relatively inexpensive in high-risk areas. Whereas average risk-based premiums in 2050 in Sweden, Ireland, and the UK are close to €400 annually per household, flat-rate premiums in Spain, France, and Belgium are approximately €13 per household. Without insurance availability, projections for France show lower floodplain population growth rates in most regions, while in the scenario with insurance availability, every region shows a higher outcome compared to the baseline (with an average of 3.5). Insurance coverage largely offsets the hazard of flood risk, and when premiums are inexpensive, the attraction of floodplain settlement more likely outweighs the costs. In countries where insurance premiums are risk-based, we generally see that population growth rates are more aligned with the baseline projection. For example, while projected population growth in the Netherlands is on average higher after introducing risk-based insurance, approximately 20% of the country's regions still show a lower outcome compared to the baseline. The introduction of flood insurance hardly affects population projections in Austria, Slovenia, and the Czech

Republic, as risk-based premiums there generally discourage settling in floodplains, similar to the scenario without insurance availability. Of the countries where premiums are risk-based, in the UK, insurance seems to drive population growth in floodplains most substantially. Although the number of UK regions where higher population growth is expected after introducing insurance availability does not change considerably (58% compared to 64% of the regions), the mean value changes from approximately 3 to 4.

A final observation in Fig. 1 is that projections for countries where premiums are risk-based in Panel C are more aligned with those in Panel B, compared to France, Belgium, and Spain (where premiums are insensitive to flood risk). This outcome is sensible considering that risk-based premiums in Panel C approach actual flood risk, as considered in Panel B. Empirical evidence that reflects this observation exists but may be mixed, as the actual implementation of accurate risk-based premiums has been limited thus far[43]. In general, the differences between Panels B and C are less striking for countries with risk-based compared to flat insurance premiums, which suggests that flood insurance premium policies can be considered a tool to reduce flood exposure and improve societal resilience against growing flood risk. The following section converts our assessment of flood exposure into future flood risk.

**Assessing future flood risk using new population projections**

Exposure is one of three elements that comprise flood risk, in addition to hazard—that is the inundation of land due to the over-topping of dykes or embankments—and vulnerability, the extent to which a flood of a certain magnitude damages exposed assets. Existing projections of exposure, obtained using population suitability maps, are insensitive to certain environmental and institutional factors that may attract or repel population growth. Including these dynamics in flood risk projections may increase their accuracy and allow for an assessment of how future flood risk may be impacted by changes in the environment or policy domain. To demonstrate this, we estimate expected annual flood damage (EAD) to residential and commercial properties using the population growth parameters obtained from Fig. 1. With this approach, we calculate flood risk while simultaneously accounting for households' adaptive responses to this flood risk, as they may choose to fully avoid potential flood damage by settling on higher ground. On the other hand, we also account for insurance policies that may limit these adaptive responses by households or even trigger risk-seeking behaviour[44].

Panel A of Fig. 2 presents the percentage change in EAD over the period 2010-2050 considering the impacts of climate change under RCP4.5 and exposure growth under SSP2, which is determined using the baseline approach as presented in Panel A of Fig. 1. Panels B to D of Fig. 2 each present the extent to which EAD may deviate from the baseline approach when assuming the population growth dynamics in relation to environmental (dis)amenities, household-level DRR, and insurance availability.

A notable observation in Panel A of Fig. 2 is that the increase in flood risk is higher in Eastern European regions compared to many regions in the West, despite the decline in exposed populations there under the baseline projection. In Poland, for example, in 2050, the EAD is expected to have doubled since 2010 to approximately €900 million, even though populations exposed to floods are expected to decline by 13%. This means that increasing flood hazard must be a strong driver of increasing flood risk in Poland and other Eastern European countries. Another reason for the higher projected growth in EAD in Eastern Europe is that modelled flood protection standards are generally lower there than in Western Europe, which means that floods are expected to occur at a higher frequency in the East. In Romania, for example, overall flood protection infrastructure is designed to withstand a 1/50 year flood, while in France, it is designed to withstand a 1/100 year flood, and in the Netherlands, a 1/1000 year flood[45]. Finally,

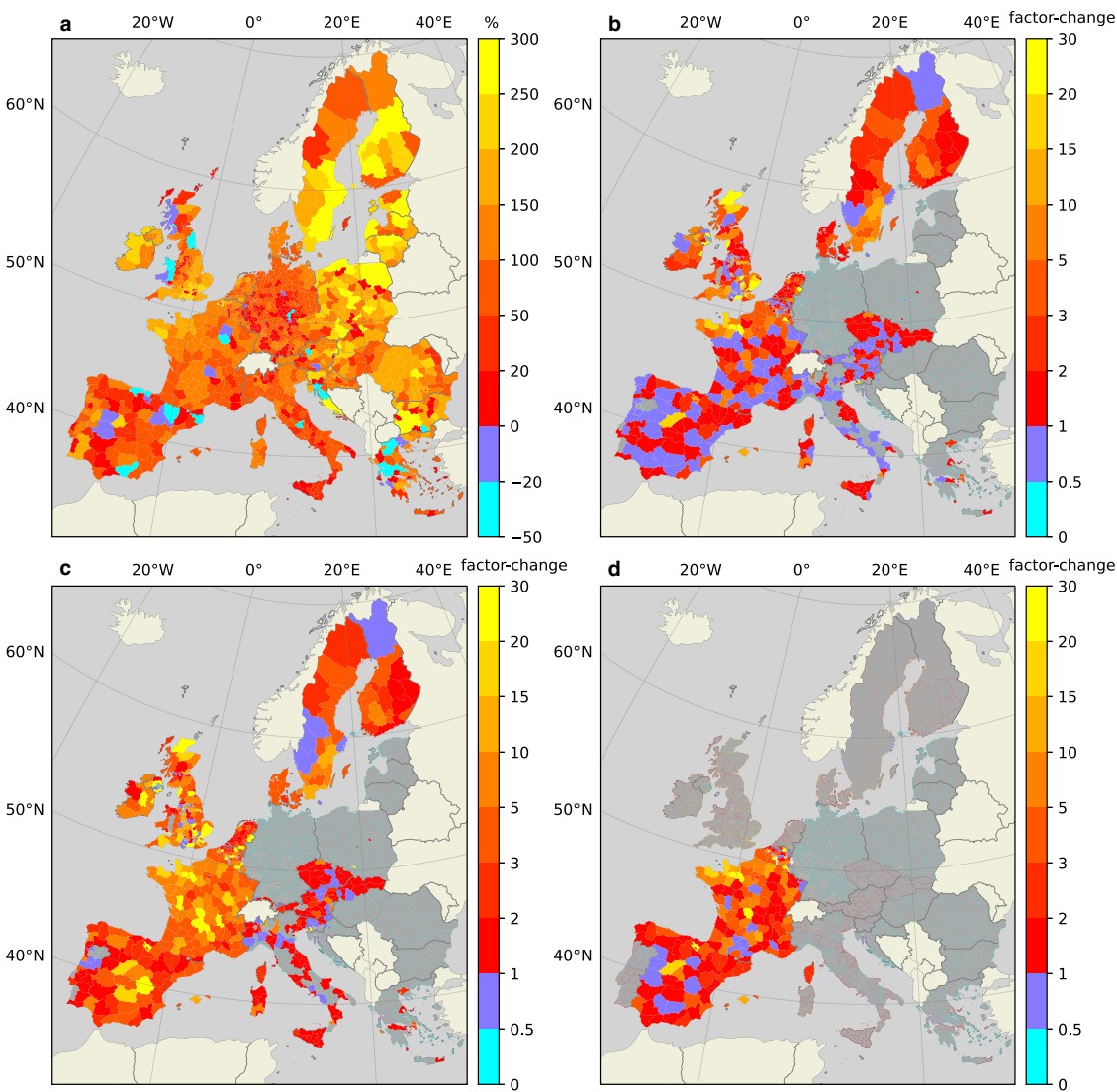

**Fig. 2 | Change in riverine flood risk from 2010 to 2050. a** The projected growth of EAD from 2010 to 2050 in percentages, under the baseline population model. Blue shades indicate a decline in EAD, while red to yellow shades indicate an increase. **b** The factor change with respect to Panel (**a**) when applying our population growth model that respects environmental (dis)amenities of floodplains and household-level adaptation. **c, d** The factor change with respect to Panel (**a**) when also considering the availability of flood insurance, where Panel (**c**) considers status-quo insurance arrangements, while Panel (**d**) displays a change to risk-based insurance premiums in countries that currently have flat-rate premiums. Blue shades in Panels (**b**–**d**) indicate lower EAD compared to Panel (**a**), while red to yellow shades indicate a higher projection. For regions depicted in grey, our method is not applicable due to declining populations across these regions (see Section "Population growth model"). Panel (**d**) also displays countries with risk-based premiums in grey.

economic growth is a likely driver of increasing EAD in Eastern Europe. This is because economic growth (expressed as GDP) considerably influences the value of exposed assets within the flood model applied in this study[2]. Although GDP is generally lower in Eastern compared to Western European countries, GDP growth rates over the period 2010–2050 are higher there[36].

Panels B and C of Fig. 2 present the deviation from the baseline projection in Panel A, corresponding to the new exposed population projections presented in Panels B and C of Fig. 1, respectively. As expected, deviations from the baseline flood risk projection are closely aligned with the underlying projections in exposed populations. That is, we see more or less the same patterns in Panels B and C of Fig. 2 and Panels B and C of Fig. 1. However, the extent of the deviations from the baseline in Fig. 2 is substantially higher (up to 30 times higher) in some regions compared to the deviations in exposed population projections (up to 10 times higher). This is because increasing flood hazard and exposure may reinforce each other and lead to larger increases in flood

risk than each of these elements alone. Considering the results presented in Panel B of Fig. 2, flood risk projections are lower than the baseline in 80% of all regions considered. However, on average over all regions, flood risk is projected to increase slightly in this scenario, which is largely driven by several regions where exposed populations are projected to increase considerably. The availability of insurance in Panel C, on the other hand, causes higher increases in EAD compared to the baseline in many regions. The most important changes in Panel C compared to Panel B can be seen in France, Belgium, and Spain, which is largely driven by higher population growth in floodplains under a flat insurance premium structure. On average, considering insurance availability, the growth in flood risk through 2050 in each of these countries is €3.9 billion, €440 million, and €1.3 billion, respectively.

Changing to risk-based premiums may be a strategy for these countries to discourage households from moving into flood-prone areas and thus limit future flood risk. In Panel D of Fig. 2, we show

results in the regular format, while we assume that current insurance systems in France, Belgium, and Spain change to become risk-based. In general, more contrasts can be seen between regions compared to Panel C, which is because local flood risk becomes a determining factor of floodplain population growth. In total, for France, Belgium, and Spain, the rise in flood risk with the availability of risk-based insurance declines by more than 50% compared to flat insurance premiums. Therefore, insurance policy design can be considered an effective tool to reduce flood risk. Although certain physical protection measures are found to be more effective at reducing flood risk, they are also highly costly. For example[46], find that improving conventional flood protection standards may reduce riverine flood risk in Europe by 70% in 2100, which will cost approximately €3.1 billion annually until then. Flood adaptation through detention areas along river systems may reduce flood risk up to 83% and cost approximately €2.6 billion per year. Concerning coastal flood risk[47], find that raising dykes reduces risk in Europe by approximately 97%, which will likely cost at least €2.5 billion per year until 2100. Several other adaptation measures, including the flood-proofing of buildings, relocation through a managed retreat strategy[46], and nature-based solutions[48], are found to be less effective than adaptation of insurance mechanisms. The exact costs of changing to a risk-based insurance system are difficult to quantify but are likely lower than the investment costs associated with the above measures. In addition to administrative costs to implement risk-based premiums, governments may need to reserve funds to assist households whose insurance coverage becomes unaffordable[19].

## Discussion
### Discussion of methods
Our simulations show that future flood exposure growth (by settlement in floodplains) is considerably influenced by developments in local flood risk, as well as the availability and price of insurance coverage. Flood risk discourages settling in a floodplain, which is aligned with[11], who find that households are sensitive to local flood risk when deciding where to settle. Insurance availability substantially increases the appeal of inhabiting flood-prone areas, particularly with higher degrees of the cross-subsidisation of flood risk. Changing flood exposure, as simulated in this study, also affects flood risk. Population growth, however, is just one of several drivers of future flood risk[5], and for this reason, it is important to put our results into perspective with other drivers; perhaps the most important of these is flood hazard. In the supplementary information it is shown that a more severe climate change scenario (RCP8.5) in combination with higher overall population growth in the EU (SSP5) causes a higher growth of EAD in many regions compared to the baseline scenario, although patterns of regional differences stay roughly the same. However, the average rise in EAD due to more severe climate change through 2050 (a 250% increase) is less than the rise in EAD due to exposure growth caused by insurance coverage (a 360% increase). This result emphasizes the importance of adaptation policy, besides policies to mitigate global warming.

Based on these findings, we can conclude that traditional climate risk assessment methods that assume exogenous exposure and vulnerability developments may inaccurately project changes in future flood risk. One reason is that households' decisions to settle in floodplains endogenously depend on flood risk, insurance coverage of this risk, and the possibility for households to apply DRR.

Our study developed a methodological framework for assessing future floodplain settlement decisions within a flood risk assessment and insurance model. Future research can use this framework as a basis and extend the analysis with climate migration decisions that include the relocations of households currently living in floodplains.

### Discussion of policy implications
Flood risk management, including the design of flood insurance policy, should already account for changing risks due to climate change, as well as those caused by changing exposure and vulnerability[49]. Policymakers must face sensitive trade-offs between maintaining an affordable flood compensation system and limiting the growth of societal losses by encouraging risk-reducing behaviour. Political myopia regarding such choices may forestall household-level adaptation, even in response to recurring floods in an area[50]. Particularly concerning economic development in floodplains, households as well as businesses make long-term decisions regarding whether to settle there or in areas less prone to flood risk. Policy choices that influence these decisions are therefore needed well in advance of climate change impacts.

As shown in this assessment, risk-based insurance is an effective tool to signal risk to policyholders and stimulate adaptation, including the discouragement of settling in floodplains. For this policy to work most effectively—that is, to target areas most significantly at risk of flooding—insurers and the government bodies responsible for premium-setting (e.g., in France or Belgium) require detailed flood risk maps. In practice, the lack of this data obstructs risk-based insurance pricing, as do other obstacles, such as the bundling of different types of risk or government-maintained caps on insurance premiums[51]. In many countries, the expectation of ad hoc government compensation —wherein (uninsured) households affected by a flood are compensated using public funds—reduces the incentive to buy flood insurance or adapt to flood hazard[20,52,53]. Although it is imperative for modern welfare states, such as EU countries, to preserve the affordability of insurance and provide assistance to households with destroyed properties, there are methods to achieve this that sustain incentives to adapt. For example, means-tested vouchers for low-income households could preserve affordability and may be combined with subsidies for installing risk-reduction measures[54].

An effective flood risk management strategy is not limited to optimising flood insurance policies; it could also include preventing floods by improving flood protection infrastructure, limiting exposure through more stringent land-use planning that for instance prohibits new construction in high risk areas, and reducing vulnerability by enforcing flood-resistant building standards. An effective adaptation strategy may require a combination of multiple elements. For example, although the physical flood protection infrastructure may effectively reduce flood risk, the higher safety standards in the protected area may trigger economic development[15,55], increasing the impact of a flood when dykes are breached. Therefore, hard measures, such as improving flood protection infrastructure, may be complemented by soft measures that trigger adaptive behaviour among households at risk of flooding.

Existing studies have assessed the impact of risk-based insurance pricing on household-level adaptation measures[19,56,57]. In this study, we add insights regarding the impact of risk-based premiums on exposure growth and the resulting trends in flood risk.

## Methods
Population development in floodplains is simulated within the existing "Dynamic Integrated Flood Insurance" (DIFI) framework[19,20,58]. This framework consists of three models that estimate flood risk, insurance premiums, and household preferences regarding several flood risk, insurance, and adaptation decisions. This study is concerned with household decision-making regarding settlement in- or outside flood-prone areas, where the primary innovation is introduced in the household decision model. Therefore, in this section we focus primarily on describing the procedure of the household decision model, while the description of the preceding models is briefly summarized. For a comprehensive explanation of all components of the DIFI-model we refer to[19]. The applied methods and flow of modelling components are depicted in Fig. 3.

The principal component of this study, the household decision model, applies several types of input data, and serves to replace a local

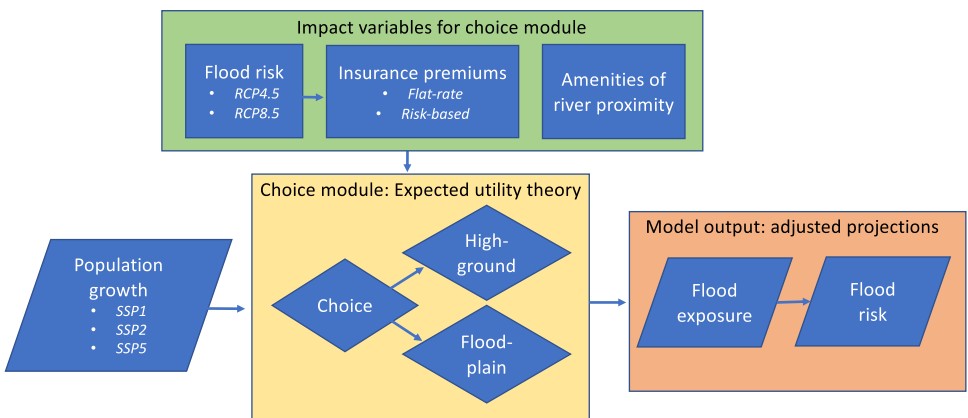

**Fig. 3 | Flowchart of model components.** References are made to sections where components are described in detail.

scale settlement suitability map as applied in[22,23]. Its purpose is to apply generic population projections and simulate where populations grow on a local or regional level. Section "Population growth model" describes how 'Shared Socio-economic Pathways' (SSPs) are applied as input for the household decision model, which comprises the choice to settle in- or outside a floodplain. In the current modelling framework, the settlement location decision that households face is based on three exogenous variables: flood risk, insurance premiums, and environmental amenities associated with river proximity.

The flood risk a household faces at a potential settlement location is based on flood hazard, which may be impacted by climate change, but also vulnerability to flooding, which may be reduced by installing DRR measures. How this is determined is explained in Section "Flood risk model".

A household may be more inclined to settle in a floodplain if the potential flood damage is compensated by insurance. The annual premium of this insurance coverage becomes an important input variable for this decision. Section "Insurance premiums" describes how insurance premiums and coverage is determined for two stylized insurance systems, which are based on flood insurance arrangements observed in EU-countries.

Rivers provide a quality that attracts populations to settle in floodplains. Such qualities include aesthetic, recreational, or broader economic values that have historically led population hubs to develop in close proximity to rivers. These river amenities are monetized for the decision framework using literature that applies hedonic pricing valuation techniques, as is described in detail in Section "Location amenities".

The choice module simulates how much of the regional population growth occurs in floodplains and how much on higher ground. With more detailed insight into population growth within floodplains, and therefore flood exposure, we can estimate regional flood risk. Section "Model output" describes the procedure of this final step.

**Population growth model**
To assess whether population growth is likely to occur in- or outside regional floodplains, we apply exogenous population growth forecasts following the "Shared Socio-economic Pathways" (SSPs), which are extensively used in research regarding future climate risks[36]. In this study, three SSP-scenarios are used that encompass a range of future developments. SSP1 corresponds to high but sustainable economic growth, which leads globally to the highest decline in population towards 2100 out of all SSPs[36]. SSP5 represents high but unsustainable global economic growth, which leads to a slightly larger global population. SSP2 depicts a continuation of historical trends in terms of social, economic, and technological development. Global population growth, in this scenario, is highest of the three that are considered.

Global population trends may, however, differ from projected developments in Europe. In Fig. 4 population growth is shown on NUTS3-level, from 2010 to 2050, for the considered scenarios. A prominent observation across all three scenarios is that populations in Eastern European regions generally decline, while those in Western European regions are expected to increase. Population decline in Eastern Europe is most notable under SSPs 1 and 2, while population growth in Western Europe is most prominent under SSP5. In several countries, such as Germany, Poland and Italy, populations are expected to decline under SSPs 1 and 2, while they increase under SSP5.

Population trends shown in Fig. 4 are acquired from the 2UP-model[22], which applies a suitability mapping approach to simulate where population growth is likely to occur. This approach, in other words, provides a tool to interpret more generic scenarios (e.g., the SSPs) on a regional or local scale. Although the 2UP-model works on a more detailed scale than the NUTS3-level shown here, the purpose of the current study is to improve upon this local analysis with innovations regarding the settlement location decision, making a particular distinction between flood-prone areas and areas safe from flooding. Therefore, population trends, as shown in Fig. 4, will feed into the household decision framework, where it is assessed whether population growth is likely to occur in floodplains or on higher ground. The household decision framework of this study only applies to population growth, which means that regions with declining populations are omitted in this analysis. It is, of course, possible that within regions populations may move from floodplains to higher ground or vice versa. However, such relocation models are likely to become highly complex, and may exceed the needs for flood risk assessments, which instead require tools to translate global population scenarios to local-scale population growth. Nevertheless, we do recognize the importance of considering relocation to- and from floodplains within regions, and we recommend future research to extend our model framework to include these dynamics.

The growth of population per region is simulated to choose to either settle in- or outside a floodplain. This choice is simulated for each new household in a region, which means that the regional population growth has to be corrected for the size of households, data for which has been obtained through Eurostat. The analysis on household-level is sensible, as flood risk assessments generally consider damage to houses and other buildings, and flood insurance also applies to buildings, not individuals. The procedure described in the following is, then, applied to each household new to a region. Each household that causes the population in the NUTS3-region to grow, therefore, is represented by a single iteration of the model.

Each simulated household receives a choice between a settlement location within a floodplain, and one outside the floodplain, on higher

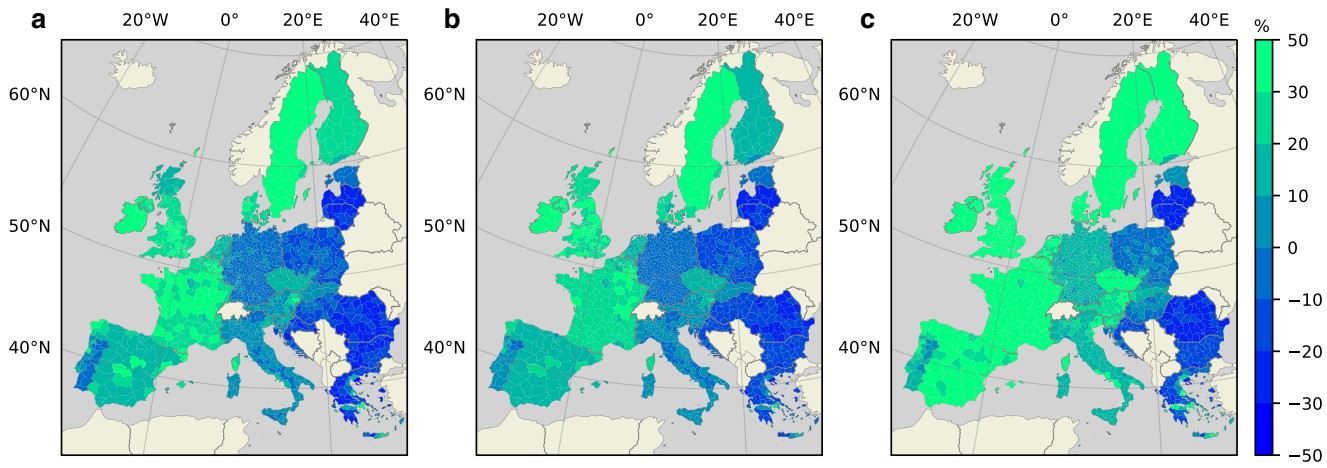

**Fig. 4 | Projected percentage population change from 2010 to 2050.** Under scenarios: SSP1 (**a**), SSP2 (**b**), and SSP5 (**c**). Blue colours represent various degrees of declining populations. Green colours represent increasing populations.

ground. The decision, then, depends on several exogenous variables, as was shown in Fig. 3. Here, we will describe in detail how each variable impacts the simulated choice for households. How the exogenous variables are derived is explained in the following sections.

The settlement location choice is summarized formally in the following equation:

$$\text{move to floodplain}_{j,i,t} \quad \text{if SEU(floodplain)}_{s,j,i,t} > \text{SEU(high ground)}_{j,i,t} \tag{1}$$

where household $j$, in region $i$, at time $t$, simulated as a single iteration of the model, chooses to move to the region's floodplain when the subjective expected utility (SEU) of settling in the floodplain is higher than the SEU of settling on the high ground. Time ($t$) is of importance because several factors that determine the settlement decision change gradually over time, most notably flood risk and the consequent insurance premium. To capture the increasing pressure of climate change, the simulation is executed in time-steps of 5 years, where for every time-step variables such as flood risk and insurance premiums are adjusted (as described in Sections "Flood risk model" and "Insurance premiums"). Because flood risk data is only available for the years 2010, 2030, 2050, and 2080, the flood risk value of the nearest year is selected for every time-step. Insurance premiums, on the other hand, can be interpolated between these years to obtain premiums for each time-step separately. The simulation to capture exposure growth from 2010 to 2050 contains 8 time-steps (and from 2010 to 2080 contains 14 time-steps). The population growth over which our downscaling simulation is applied for every time-step is determined by dividing the total population growth over the period 2010 to 2050 (or 2010 to 2080) by 8 (or 14).

The SEU associated with floodplain settlement is subject to subscript $s$, which represents the household's planned strategy when settling in the floodplain. Households may find settling in a floodplain unattractive compared to the high ground when potential flood damage is not compensated. However, obtaining insurance coverage or applying a DRR measure may change this outcome. The floodplain SEU-function, therefore, differs for each strategy $s$, which is either; no action, obtain insurance coverage, or apply a DRR measure. The household will choose the floodplain settlement location if the SEU associated with any one of these strategies is higher than the SEU of settling on the high ground.

Equations (2) and (3) present the expected utility functions for settling in the floodplain and on the high ground respectively.

$$
\begin{aligned}
\text{SEU(floodplain)}_{s1,j,i,t} &= \beta_j p_i U(W_{j,i,t} - \gamma_j L_{j,i,t} + A_{j,i,t}) \\
&\quad + (1 - \beta_j p_i) U(W_{j,i,t} + A_{j,i,t}) \\
\text{SEU(floodplain)}_{s2,j,i,t} &= \beta_j p_i U(W_{j,i,t} - \pi_{i,t} - \alpha\gamma_j L_{j,i,t} + A_{j,i,t}) \\
&\quad + (1 - \beta_j p_i) U(W_{j,i,t} - \pi_{i,t} + Aj,i,t) \\
\text{SEU(floodplain)}_{s3,j,i,t} &= \beta_j p_i U(W_{j,i,t} - C - (1-\delta)\gamma_j L_{j,i,t} + A_{j,i,t}) \\
&\quad + (1 - \beta_j p_i) U(W_{j,i,t} - C + A_{j,i,t})
\end{aligned}
\tag{2}
$$

Equation (2) presents the SEU of floodplain settlement for the three strategies separately, which are from top to bottom: no action (s1), obtain insurance coverage (s2), and apply a DRR measure (s3). The structure of each SEU-function is similar, with only additional costs and reduced flood losses in s2 and s3.

Fundamentally, the floodplain SEU-function considers two states of wealth; one where a flood occurs with probability $p_i$, and one where no flood occurs $(1 - p_i)$. In the state where a flood occurs, the household's wealth endowment $W_{j,i,t}$, increased by an amenity value generated by river proximity $A_{j,i,t}$, is subtracted by flood losses $L_{j,i,t}$. In the state where no flood occurs, the household maintains it's initial wealth in addition to the amenity value. Both states of wealth are transformed using a logarithmic utility function $U$, which exhibits constant relative risk aversion, and is commonly used to model human decision-making under risk[59]. The flood probability $p_i$ is determined on a regional level, based on flood protection standards acquired from the FLOPROS database[45]. Based on this data, regional flood protection infrastructure that protects against a 1/100-year flood, for example, converts into a $p_i$ of 0.01 in our SEU framework. The wealth endowment $W_{j,i,t}$ is the total financial means a household has access to, which is assumed a fixed proportion of income as proposed by Eurostat[60], data for which is obtained from Eurostat on NUTS3-level. Future values of $W_{j,i,t}$ develop in accordance with GDP projections, following the SSP scenario[36]. Expected flood damage $L_{j,i,t}$ is the outcome of applying the flood probability ($p_i$) to the probability-impact curve (see Section "Flood risk model"). Amenities $A_{j,i,t}$ associated with settling in the proximity of a river is approximated using values obtained from literature that apply hedonic pricing methods, which generally concludes that amenities provided by a river decline with increasing distance from the river. The exact approach used for this is described in Section "Location amenities".

The flood probability $p_i$ and associated flood damage $L_{j,i,t}$ are adjusted to represent perceived risk, which is done in order to

circumvent the strong rationality assumptions that underlie expected utility theory. Particularly concerning low-probability-high-impact risks such as flooding it is found that decision-making by individuals does not follow rationality assumptions laid out in traditional economic theory. For example, even though risk-averse individuals should desire an insurance mechanism against disasters that can financially ruin them, individuals generally disregard low-probability flood risk and choose not to insure[61–63]. Both the flood probability misperception parameter $\beta_j$ and the flood impact misperception parameter $\gamma_j$ are based on empirical findings in[62], which observe that flood probabilities are found to be generally overestimated, whereas impacts are more likely to be undervalued. More specifically, $\beta_j$ is drawn randomly from a normal distribution with $\mu = 5$ and $\sigma = 0.5$, whereas $\gamma_j$ is drawn randomly from a normal distribution with $\mu = 0.8$ and $\sigma = 0.2$.

When considering the floodplain settlement location, households have the option to increase their expected utility by insuring against flood damage, or by reducing flood risk through DRR-applications that flood-proof their building. Strategies s2 and s3 in Equation (2) represent how these SEUs are derived respectively. Regarding the option to insure, wealth $W_{j,i,t}$ is subtracted by the insurance premium $\pi_{i,t}$ regardless of whether a flood occurs or not. The insurance premium a household faces is dependent on the national flood insurance system, which means it can be either reflective of local flood risk conditions, or based on national average flood risk (see Section "Insurance premiums" for details about the simulation of insurance premiums and the allocation of European countries to stylized insurance systems). Insuring flood risk means that perceived flood losses ($\gamma_j L_{j,i,t}$) are reduced by the level of insurance coverage, meaning that insured households only pay a deductible in the case of a damaging flood. The deductible $\alpha$ equals 15% of flood damage, which is based on a review of European flood insurance systems[64]. A final note on the expected utility with insurance is that in some countries insurance coverage is mandatory for all homeowners (i.e., France, Belgium, and Spain), which, for our simulation with insurance availability, implies that households in these countries can only consider $SEU_{s2}$.

The option to settle in a floodplain and reduce potential flood damage by applying a DRR-measure requires an investment cost $C$, which applies regardless of a flood occurrence. The DRR-measure reduces perceived flood losses $\gamma_j L_{j,i,t}$ by an effectiveness-parameter $\delta$. This study considers two types of DRR-measures—dry- and wet flood-proofing—which vary in terms of costs and effectiveness at reducing flood damage, estimates for which are obtained from[65]. Dry flood-proofing implies barriers that prevent flood water from entering a property, which is estimated to cost ($C$) €471 and reduce flood risk ($\delta$) by 13%. Dry flood-proofing measures are effective at preventing damage up to a certain level of inundation, after which they are overtopped and unable to prevent damage at all. On the other hand, wet flood-proofing consists of measures to reduce flood damage when water enters a property, such as applying water-resistant floors and building materials, and fitting electrical appliances on higher floors. By allowing water to enter a building, it is less likely to collapse due to hydrostatic pressure from rising water levels outside. The costs ($C$) of wet flood-proofing are estimated at €2389 and it is considered to reduce flood damage ($\delta$) by 25%.

Finally, the amenities of river proximity, reflected through housing prices, do not express a yearly value, but rather the benefits of a location over the time a household expects to reside in a property. Therefore, within the SEU-framework it is incorrect to compare a long-term benefit with annual flood or insurance costs. For this reason, the insurance premium ($\pi_{i,t}$) in Equation (2) is an aggregation of the yearly premium over the expected time of residence, which is assumed to be 15 years (based on the average length of residence in a single home in the UK). Although, theoretically, future premium payments need to be discounted when expressed in current terms, including a discounting procedure does not change the outcome notably considering the small

confined period of 15 years, as was tested using the model. Hence, we chose not to apply discounting to prevent making the model unnecessarily complex. Moreover, the short individual planning horizon of 15 years already captures time preferences to a certain degree. A residence time of 15 years also means that a flood is more likely to occur compared to a single year. For this reason, the perceived annual flood probability $\beta_j p_i$ is multiplied by 15 in Equation (2).

Outside the floodplain, on higher ground, there is no riverine flood risk, meaning the SEU associated with settling there only considers wealth and amenities, depending on the distance from the river. In countries where flood insurance maintains risk-based premiums, households that settle on higher ground do not have to pay an insurance premium, as there is no flood risk. However, in countries where insurance pricing is insensitive to the risk of individual households, and, therefore, flood risk is cross-subsidized between households in high- and low-risk areas, households located on higher ground will have to pay a premium. The SEU of settling on higher ground can therefore, depending on the national insurance arrangement, be given by the following:

$$SEU(\text{high ground})_{j,i,t} = U(W_{j,i,t} - \pi_{i,t} + A_{j,i,t}) \tag{3}$$

## Flood risk model

The flood risk model produces two important outputs on NUTS3-level, that are used for further analyses. Firstly, it generates a flood probability-impact curve, which is applied in the expected utility framework to determine flood losses associated with a certain probability. Secondly, it estimates Expected Annual flood Damage (EAD), which is used to determine flood insurance premiums.

To estimate fluvial flood impacts and construct the flood probability-impact curve, we employ the GLOFRIS risk assessment framework as applied by[2] and[5]. This framework integrates three crucial components: flood hazard, exposure, and vulnerability. Flood hazard is evaluated using inundation maps, which outline the extent of flooding at a spatial resolution of 30' × 30'. Simulations are conducted for various return periods (2-, 5-, 10-, 25-, 50-, 100-, 250-, 500-, and 1000-years), representing different flood probabilities across four time points: 2010, 2030, 2050, and 2080. Future flood hazard is simulated using a meteorological model, which requires an assumption regarding the level of greenhouse gas accumulation in the atmosphere. The Global Circulation Model HadGEM[66] serves as the meteorological model for our simulation and applies levels of greenhouse gas accumulation according to several Representative Concentration Pathways (RCP). The baseline analysis (presented in the main text) applies RCP4.5, which represents a future where the 2 °C limit set by the Paris Climate Agreement is met[35,36]. As a sensitivity analysis, also RCP8.5, a more pessimistic scenario[67], is applied and presented in the Supplementary Information.

In order to assess the potential physical damage caused by inundation to the built environment, it is essential to consider exposure to flooding. In this study, we define floodplains as regions that could be affected by a 100-year flood, corresponding to a 1% annual flood probability. To estimate flood damages within these areas, it is crucial to identify and evaluate the built environment present there. Because this study concerns exposure developments due to population growth, we limit the assessment of flood risk to residential exposure, which is assumed to make up 75% of the total built environment[6]. To acquire this data, we utilize the 2UP-model[22], which incorporates socio-economic forecasts based on the Shared Socio-economic Pathways (SSPs). The baseline scenario, presented in the main text, adopts SSP2, representing a continuation of current socio-economic trends[35,36]. Additionally, a sensitivity analysis is conducted using two alternative scenarios: SSP1, which reflects rapid sustainable socio-economic growth[7], and SSP5, which represents rapid unsustainable socio-

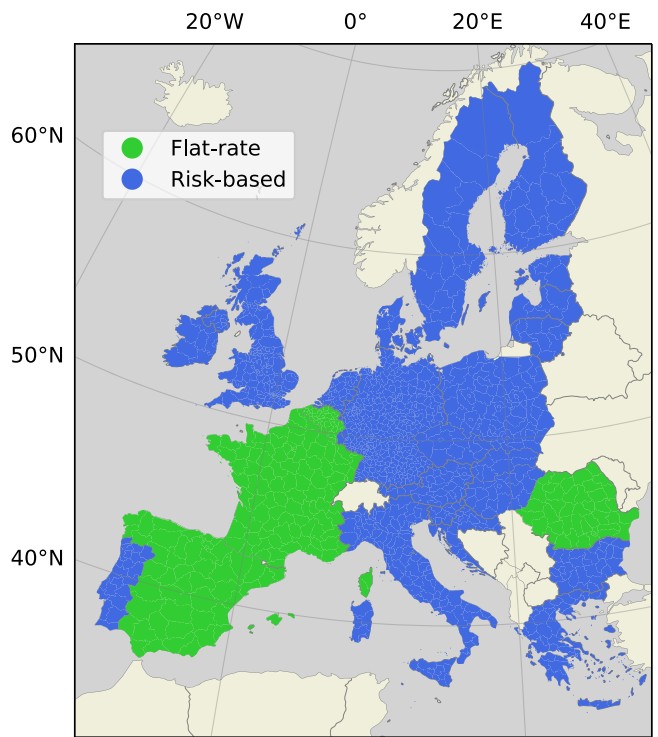

**Fig. 5 | Allocation of countries to stylized insurance arrangements.** "Risk-based" indicates that insurance premiums reflect the risk of individual policyholders to an extent. "Flat-rate" means that premiums are insensitive to the risk of individual policyholders.

economic growth[7]. These scenarios allow for a comprehensive examination of the potential flood damages in different socio-economic contexts.

Vulnerability is considered by utilizing depth-damage curves that establish the relationship between flood depth and anticipated damage[68]. The flood probability-impact curve is constructed by combining flood hazard, exposure, and vulnerability.

By applying the method explained above, a flood probability-impact curve can be constructed. EAD can be estimated based on the surface of this curve, while also considering that certain water-levels do not cause inundation due to flood protection infrastructure. These flood protection standards, also defined in terms of return-period, are obtained from the FLOPROS database[45]. EAD, then, is determined by the residual risk, which is the section of the probability-impact curve that exceeds the local protection standard. We refer to[48] for a more detailed explanation of this process.

**Insurance premiums**
Flood risk and population data is used to determine flood insurance premiums in the DIFI-model. We discern two types of flood insurance for this study; one where premiums reflect the flood risk faced by individual households, and one where premiums reflect the flood risk aggregated to country-level. These types of insurance reflect distinct policy choices, where risk-based premiums can be associated with economic efficiency, and flat-rate premiums reflect a solidarity principle. Although, in practice, there are variations regarding the extent that either policy is implemented, the main objective of our insurance analysis is to assess the impact of a key difference in insurance policies on settlement choices and flood risk. A requirement for the calculation of accurate risk-based insurance premiums is detailed flood risk maps, which often proves to limit the implementation of risk-based premiums[43,51]. In our assessment, the flood insurance system is administered on country-level, and countries are allocated to either one of

these stylized systems based on a review of European insurance arrangements (see the appendix in[20] for this review). The allocation of European countries to the stylized insurance arrangements is presented in Fig. 5.

The process of calculating premiums for the two insurance systems can be described intuitively. The flat-rate premium, characterized by full cross-subsidization of risk between high- and low-risk households, is determined by dividing the aggregated riverine flood risk (EAD) in a country by the total number of households residing in the country. Premium-setting in this insurance system, as applied in France, Spain, and Belgium, is organized centrally by governments. Insurance coverage can still be provided by private insurers, as in Belgium and France, but is sometimes provided by governments themselves, such as in Spain.

Risk-based premiums are estimated for NUTS3-regions by spreading EAD amongst the number of households located in 100-year floodplains. Households located outside the floodplain pay no premium, while those located within a floodplain pay a premium that reflects the average risk within the floodplain. Therefore, in our calculation of risk-based premiums, there remains a degree of cross-subsidization between those most severely at risk, and those that face somewhat lower flood risk. This estimation process is more representative of how risk-based premiums are currently determined, which is done on fairly coarse levels[51].

A detailed formal description of how the risk-based and flat-rate flood insurance premiums are calculated is given in[69].

**Location amenities**
Amenities associated with river proximity decline over space, evidence for which will be presented in this section. Therefore, as the floodplain directly encloses the river, amenity values outside the floodplain will always be lower than inside. In order to assess the amenity values in our simulation, we randomly select two potential locations for a household to settle, one inside the floodplain, and one outside the floodplain, which we will refer to as the high ground. The benefits of each location are estimated using a function of amenities where values decline over distance.

The distance of a potential settlement location from the river is approximated using flood inundation maps derived from the GLOFRIS model, which is able to compute the surface area of riverine floodplains for each NUTS3-region. Floodplains, here, are estimated by simulating the excess surface water resulting from riverine flooding, which is assessed using a hydrological model[70]. As the reader may imagine, selecting the two potential settlement locations and determining their respective distance from a river is a complex task on a map where floodplains take all kinds of shapes. To simplify this procedure, the surface area of a NUTS3-region is transformed from a 2-dimensional map to a 1-dimensional line, where the floodplain makes up the segment closest to the river and the remainder consists of high ground. To realize this, a conversion is carried out to obtain the relative size of the floodplain to the total area of the NUTS3-region, the result of which is shown in Fig. 6. The two potential settlement locations, then, are selected in terms of the relative size of the floodplain. For example, in regions located in the Rhine and Po river deltas, approximately 80% of which are made up of floodplains, the potential floodplain settlement location is randomly selected within this area, whereas the potential high ground settlement location is randomly selected in the remaining 20% of the region's area that is furthest away from the river. In many other NUTS3-regions, floodplains make up less than 5% of the total area, which means the potential floodplain settlement location is randomly selected within this small area surrounding the river, whereas the high ground location can be anywhere in the remaining 95% of the region.

After selecting the two potential settlement locations, the next step is to estimate the value of river amenities for both. To do this, we

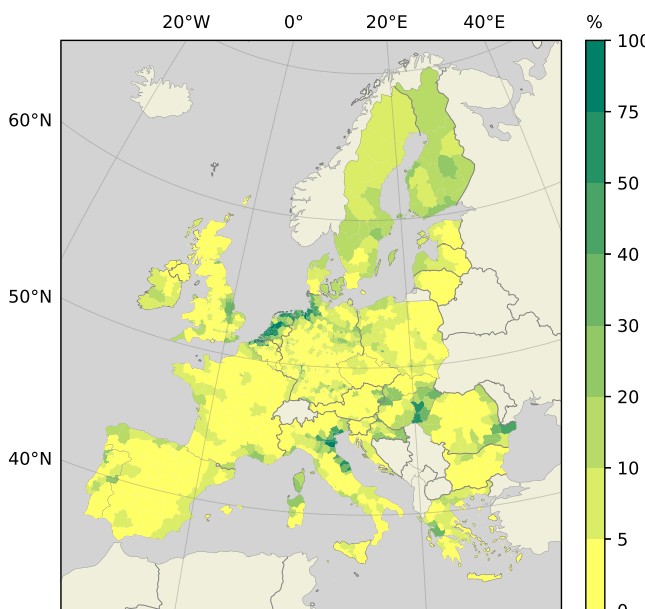

**Fig. 6 | The size of riverine floodplains as a percentage of the total NUTS3-area.** Dark green colours indicate that a region predominantly consists of floodplains. Regions shown in light green colours have only a small share of their territories consisting of floodplains, or have no floodplains at all.

apply insights from hedonic pricing literature, which is a valuation technique where property market prices are used to determine the value of a variety of location (dis)amenities associated with nearby environmental characteristics and residential facilities. For our analysis we are interested in the hedonic pricing values of river landscapes, which may include aesthetic values, but also opportunities for recreation[71]. The Hedonic Pricing Method (HPM) assumes that environmental characteristics, both positive and negative, through forces of supply and demand affect house prices. To unravel the value of such characteristics, a HPM-study seeks to compare the price of a house in the vicinity of certain (dis-)amenities with a comparable house further away from these (dis-)amenities.

Studies on the value of river amenities may be subject to local characteristics of a specific ecosystem, such as the water-quality[41] or the state of the river's embankment (e.g. whether natural or endyked). For example, the results of a HPM-study of a particularly pristine section of a river in a high-income region may not be generalized to other sections of the river, let alone to other rivers. Also the size of the river, or the type of water flow, impacts the value of amenities obtained through HPM.[72], for example, find that there are significant differences in the valuation of streams, rivers and bayous in the US Gulf coast. While the proximity of rivers and streams have a positive impact on amenity values, the opposite is found for bayous (a type of estuary). Therefore, using HPM studies to obtain a value for river proximity on the coarse multinational scale necessary for this study, we need to choose a functional form of the impact of river proximity on house prices that may be generalized across the continent.

HPM is used in existing literature to value a broad range of environmental amenities. For example[42], apply over a million house transactions across the UK to value amenities provided by various landscapes, such as domestic gardens, green spaces and water. They are able to conclude for each of these that a percentage point increase of their presence in a region causes a significant rise in house prices within that region. Moreover, they find that a 1km increase in distance from a river causes house values to decline by 0.93%.[13] collected findings from 46 articles applying HPM, resulting in 84 effects measured, and summarize the results for multiple categories of

landscapes, such as forests, wetlands and agriculture. Roughly half of all observations show positive and statistically significant effects of natural amenities on house prices[71]. Provide an overview of HPM-literature that focus on amenities provided specifically by waterways, such as rivers, streams and canals. Results that are summarized in this study vary substantially depending on the type of waterway, but also whether the waterway is in an urban or rural area. The strongest amenity values of rivers are found in urban areas, and the impact of river proximity is particularly high in studies that consider houses with a river view. For example[73], find that bordering the Farmington River in Connecticut US accounts for 42% of the land value, or $168/ft, which diminishes to $3.76/ft for plots 1 mile away from the river. Similarly[74], find that for houses located 1km away from the Murray-Darling river in South-Australia, decreasing the distance by 0.5km increases the house price by AU$245.000. It should be noted that both of these are examples of HPM-studies in specific wealthy areas, close to urban centres. A larger spatial scale in HPM-studies generally reduces the impacts of amenities on property values compared to the examples cited above. For example[75], analyze almost 25.000 property transactions in Mineapolis-St. Paul, Minnesota, and find that sales prices increase on average 0.027% for every 1% decrease in the distance from the nearest river. For houses located within 10% of the maximum distance from a river, approximately 2km, this marginal effect is almost 0.1%, while there seizes to be a discernible effect after 50% distance from a river (approx. 10km).

The final amenity value of river proximity used in the model ($A_{j,i,t}$) can be expressed by the following piecewise linear function:

$$A(x_j)_{i,t} = GDP_{i,t} * \theta_i^* \begin{cases} 4000 - 150x_j & x_j < 10 \\ 2500 - 100(x_j - 10) & 10 < x_j < 25 \\ 1000 - 40(x_j - 25) & 25 < x_j < 50 \\ 0 & x_j > 50 \end{cases} \quad (4)$$

where $x_j$ represents the distance of a household's potential settlement location from the river. It can be seen that the marginal decrease in amenity value declines with distance from the river, which is aligned with empirical findings[74,75]. Because average house values differ considerably between European countries, we correct amenity values to the deviation of average national housing prices from the European average, which is represented by $\theta_i$, and data for which is obtained from Eurostat[76]. Finally, since the value of amenities is determined based on property prices, this should also change accordingly with expected developments in property prices. For this purpose, the amenity value is adjusted to projected changes in future regional GDP, which is standardized to 1 for the baseline year (2010) and expressed relative to 2010 for for each time-step until 2080. Data for this GDP ratio is obtained from IIASA's SSP database[36], and conforms to the SSP-scenario's used for this study.

The final amenity value of river proximity approaches values found by[75]. An important reason for this choice is that our simulation is applied on a large geographical scale, which comprises a diverse range of landscapes, wealth, and cultures. While for some specifically wealthy locations, or rivers of particularly pristine beauty, the resulting hedonic value may be an underestimation, our choice is appropriate concerning the spatial scale of this analysis. For more detailed local-scale analyses it will be useful to apply hedonic pricing values that are representative of the specific location.

We conducted an analysis to test the sensitivity of the simulation to the value of river proximity, which is presented in the Supplementary Information.

### Model output

This study produces two types of results: flood exposure projections, and estimations of future flood risk (see Fig. 3). Firstly, we are interested in how much the population changes in floodplains. If it is

attractive to reside in floodplains, for instance due to positive amenities and low flood risk or cheap insurance coverage, then population growth within floodplains may be higher compared to projections that do not consider these drivers. Population growth in floodplains directly increases flood exposure, as this requires new residences to be built.

After simulating population growth under various insurance scenarios and adaptation options, it is possible to project how flood risk changes respectively under these scenarios of flood exposure development. To do so, we return to the flood risk model and adjust the flood damage estimates per return period, which are used to simulate EAD, based on the modified population exposed per NUTS3-region. A straightforward calculation is made where the flood damage estimates for 2050 (or 2080) are adjusted proportionally to the floodplain population projections. For example, if population growth in our projection is expected to double in a region compared to the baseline projection, the damage estimates for each of the 9 considered return periods is also doubled in that region. After this, the procedure described in Section "Flood risk model" is followed to obtain EAD under the different insurance and adaptation scenarios.

### Reporting summary
Further information on research design is available in the Nature Portfolio Reporting Summary linked to this article.

## Data availability
Flood damage estimates and the floodplain surface area per NUTS3-region are obtained from the GLOFRIS cascade model[2,5]. Data on flood protection standards are acquired from the FLOPROS database[45]. Socio-economic data is obtained from IIASA's SSP-database[36]. Baseline population growth projections apply data from the 2UP-model[22]. Income data is obtained from Eurostat and transformed to a log-normal distribution in[19]. Amenity values are rescaled to country-level housing prices obtained from Eurostat[76]. The data used to execute the model developed in this study is available in Zenodo repositories. Data related to flood risk, obtained from the GLOFRIS model, is accessible with the link: https://doi.org/10.5281/zenodo.10033587. The remaining data used to execute the model is published together with the simulation code, and is accessible using the link: https://doi.org/10.5281/zenodo.8187319. Raw data underlying the figures of this study is available on Figshare using the following link: https://doi.org/10.6084/m9.figshare.23798667. Figures presented in this paper are constructed in the Python programming language, using the packages Cartopy (https://doi.org/10.5281/zenodo.8216315) for background map features, and Matplotlib (https://doi.org/10.5281/zenodo.10059757) for visualizing data.

## Code availability
The code of the main innovation of this study is published on Zenodo, and may be accessed with the following link: https://doi.org/10.5281/zenodo.8187319. Code of earlier versions of the insurance model is available upon request.

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

## Acknowledgements
Research and preparation of this manuscript done by M.T., W.J.W.B, and J.C.J.H.A. was funded by the EU-ERC COASTMOVE project (no. 884442 to J.C.J.H. Aerts) and the EU Horizon project ACCREU (no. 101081358 to W.J.W. Botzen). T.T. is funded by the European Union's Horizon 2020 MYRIAD-EU project; Grant Agreement No. 101003276. This work used the Dutch national e-infrastructure with the support of the SURF Cooperative using Grant No. EINF-4493.

## Author contributions
The work was conceptualized by M.T. and W.J.W.B. The methodology was developed by M.T., W.J.W.B., and T.T. Formal analysis was undertaken by M.T. and T.T. Visualization was by M.T. and T.T. Supervision and funding acquisition were by J.C.J.H.A. and W.J.W.B. All authors assisted in writing the manuscript.

## Competing interests
The authors declare no competing interests.
