## [Peer Review File · Nature Communications]

Flood insurance is a driver of population growth
in European floodplainsREVIEWER COMMENTS

Reviewer #1 (Remarks to the Author):

The paper presents an innovative method to project future flood risk under climate change scenarios that simulate population growth in floodplains, considering how amenities associated with river proximity, flood risk, adaptation options and insurance against risk influence settlement location decision. Obtained results are innovative and significant both from a scientific point of view and for decision makers involved in risk mitigation. I congratulate with authors for the great job. Result interpretation is mostly in agreement with model results. I do not have significant comments on the main text, only suggestions on how to improve its clarity (see specific comments in the manuscript). My main concern is about the lack of clarity on the implemented methodology that in some cases hampers the interpretation of results (for instance, on the effect of adaptation options or environmental amenities). There are parts of the methodology that are not totally clear, especially regarding how flood damage is evaluated, considering also the effect of adaptation options, or environmental amenities. I had also difficulties with the adopted symbolisms. Finally, I think more details should be supplied regarding model calibration and validation. Figures must be improved as mistakes are present and, sometimes, they are not self-explanatory. Please, refer to my specific comments in the attached annotated manuscript for further details. The manuscript is worthy for publication after unclear points will be addressed.

Reviewer #2 (Remarks to the Author):

The paper presents a new methodology that takes into account human's behaviour changes in responding to flood insurance and environmental amenities in estimating population exposure to floods. It is innovative but I do have some reservation (see my comment 1 and 7). I recommend its publication after major revision. My comments are provided as follows.

1. I find the maps shown in NUTS3-level misleading. The study focuses on river flooding but the spatial aggregation covers the coastal areas as well. The latter has very different flood exposure due to coastal flooding or combined river/coastal flooding. It is not clear if the population exposure obtained by using the 2UP model is also for river flooding and if the population data used do not include population in coastal areas. Fig 1 shows the key results within the main manuscript, but it is too small to see the spatial differences. It needs to be enlarged and ideally the coastal areas should be excluded. The discussion in the manuscript always refers to countries, and as such the country boundaries should be visible on the map.
2. It is not clear where the 100-year floodplains are located. The authors referred to it throughout the manuscript, for example line 119-122.
3. One can hardly see any floodplains in the maps but rather entire countries or regions without much difference in the colours.
4. It was not explained in the manuscript why the 100-year return period was used for the analysis. Surely the more frequent floods such as 50-year would be of more interest unless the authors assume the protection standards are higher than 50-year all over Europe. The second issue is the 100-year flood extrapolated using shorter time slices is not robust.
5. Throughout the paper, the term of 'risk' and 'hazard' are used interchangeably without making distinction. The authors need to distinguish between 'flood risk' and 'flood hazard' and define the terms of hazard, vulnerability and exposure that make up 'flood risk'. The definitions in line 277-280 are not accurate. Hazard is not "inundation of land due to the over-topping of dykes or embankment."

Vulnerability is not just “the extent that exposed assets are damaged by a flood of a certain magnitude.”

6. One important aspect missing in this study is government intervention in flood risk management. There are flood zones that are prohibited from development or need planning permission. For example, the Environment Agency in England provides a flood zone map and they have specific regulations on what types of building and what kind of planning permission would be required in a certain flood zone. This study seems to assume the population settlement is completely free from government regulations and planning policies. The equation A1 in Section A.1.2 is rather simplistic assuming settlement decision is completely made by individuals without government regulations. Insurance premium could put constraints on floodplain settlement, but simply increasing the premium is not a solution as it would disadvantage certain households and individuals. The message would be misleading if we simply increase insurance premiums with the aim of reducing exposure. I would like to see discussion on the impacts or effects of government intervention in addition to completely insurance policy driven analysis.

7. Further to my comment 1, Figure A4 has major flaws with regards to the floodplains. If the study focuses on river flooding only, it is incorrect to see 100% percentage of floodplain areas along the coasts. For example, the Rhine, Po river deltas, the Broadlands in east England etc. These areas are dominated by coastal flooding rather than river flooding.

8. Fig A1: Flood risk appears in “Impact variables for choice module”, but also in “Model output”. It is very confusing. I think the Flood risk in “Impact variables for choice module” should actually be “Flood hazard” or probabilities, and the Flood risk in “Model output” should actually be “Flood damage” or EAD.

9. Section A.2 Flood risk model

The heading should change to ‘Flood damage’ or ‘Flood loss’ model because that you describe here is the method obtaining the flood damage and losses.

The study uses 100-year return floodplain to estimate population exposure, but the method describe here uses a range of return periods from 2- to up to 1000-year. This is rather confusing. Is the annual damaged derived for 100-year return period only or all the return periods beyond 100-year?

10. The manuscript needs English proof reading to make it easier to read. There are sentences that are formulated in strange ways and difficult to understand. Some sentences are too long and can be broken down to shorter sentences.

Technical corrections:

Line 74: the most effective method to limit societal exposure

Suggest change to “the most effective method to limit population exposure” as it is better to stick to your terminology.

Line 167-174

Instead of using ‘flood risk’ here, it should be ‘flood losses’ or ‘flood damage’. It is also not clear if this is annual flood damage of 100-year return flood or flood damage of any unprotected magnitude.

Line 216, This person more likely prefers a lower

Change to ‘Such people more likely prefer’

Line 289: fully avoid potential flood damage by moving out of harm’s way

What do you mean by “harm’s way”. Please consider rephrasing here.

Line 334: the growth in flood risk
Should be 'flood damage'

Line 452: remove ', then,'

Line 571: flood risk (L)
This should be flood losses (L). In line 579, "flood damage (L)" is used. Please be consistent with your terminology.

Line 618: 15 years
Please explain why using 15 years in your model.

Line 839: future flood risk
Should change to 'future flood damage'

Line 856: regular procedure is followed
What is the regular procedure?

Reviewer #3 (Remarks to the Author):

The manuscript presents an analysis of future river flood exposure and risk scenarios in Europe, based on a modelling framework that can simulate population growth in floodplains considering environmental and institutional factors.

The methodology applied covers a relevant aspect of flood risk management which has not been explored previously by continental-scale models, at least to my knowledge. The findings are also of interest, as they shed light on some (unintended) linkages between risk insurance policies and changes in flood exposure and vulnerability. Thus, work might be an important contribution to the scientific literature in the field.

However, some issues need to be addressed before reaching a conclusion about the publication of the manuscript:

- The actual degree of novelty of the modelling framework and of the outcomes is not completely clear to me. The study builds on a number of already-existing models and studies (GLOFRIS for flood risk analysis, 2UP for general population growth, DIFI for insurance modelling) by adding new components for population growth and flood-proofing of buildings. It is not clear to me whether some results, such as EAD estimations, are taken from previous works from the same Authors or are new, and how much of the previous studies by the Authors overlap with the present work. I would invite the authors to briefly describe their past works in the topic, as well as other relevant literature works (see also the following point), and then specify what are the innovations brought by their new analysis.

- The modelling framework focuses on the role of individual or household-based choices in shaping flood exposure and vulnerability, which is perfectly fine. However, the overall discussion of results seem to overstate the relevance of Authors' findings while overlooking the role of flood risk control measures driven by government policies, such as structural measures, land-use planning and so on. Historically, structural flood control measures (river dikes, dams, detention areas etc) have been determinant in shaping floodplain development (Haer et al., 2016; Di Baldassarre et al., 2018), whereas government policies can strongly influence flood exposure by limiting or excluding further developments in flood-prone areas, so individual decision are only a part of this framework. In addition, a number of recent studies evaluated future risk scenarios resulting from a top-down implementation of these and other measures (Ward et al., 2017, Vousdoukas et al., 2020; Tiggeloven et al., 2020, Dottori et al., 2023, with apologies for the self-citation). My impression is that the role of

these measures in shaping flood risk can be , at least, as important as the measures analysed in the present manuscript. Therefore, I would recommend including all these aspects in the introduction and in the discussion of results, in order to put the authors' findings in the right perspective.

- In the main text there is no explanation on why only some EU countries/NUTS3 regions are represented in Panels 1B-1C and 2B-2C. My understanding from Section A1.1.1 (L500-501) is that population growth in floodplains is evaluated only in NUTS3 regions with overall increasing growing population trends, but this seems in contrast with values in Figure A2-A (where Italy seems to have positive trends, for instance) and with the projected increases in few areas in eastern countries (e.g. Bucharest, Warsaw, Athens). Could you please explain this point?

- I understand that a formal validation of the model is probably not possible. Still, I would ask the Authors to describe how their results compare with the study by Tellman et al after the calibration (in particular, if the model was able to reproduce observed dynamics)

Minor points

Title: my impression is that now the title suggests that insurance is the main driver of population growth in floodplains, which is not the case. Perhaps a more nuanced wording would be more appropriate

You should clearly specify in the title, abstract and introduction that the work is about river flooding and Europe. Similarly, along the text you should mention whether your statements refer to floods in general (either river, coastal, flash or pluvial) or river floods only

L93 I would briefly mention here the SSP and RCP scenarios that have been considered in the study.

Figure 1: Legends are not fully clear in this Figure, I suggest using the same definitions as in Figure 2. Also, for panel A, using a colour palette with different colours for negative - positive change (as in panels B-C), would make the map more readable

L236: "Interestingly, some of these are regions where projections without insurance availability..."
Could you list some of these regions here?

L240-243: "... population growth projections in floodplains are consistently and considerably higher in France and Belgium, where premiums are relatively inexpensive in high risk areas." Could you provide an average monetary value for risk-insensitive and risk-based premiums as estimated by the model?

-Section 2.4: After having read section A.5, my understanding is that you considered damage to residential and commercial areas to calculate EAD, is it correct? If yes, please specify this in the main text.

Figure 2: In panel A, using a colour palette with different colours for negative - positive change (as in panels B-C-D), would make the map more readable. Also, can you please explain how "factor" is calculated for panels B,C,D?

L304-309: Did you consider also projections of economic growth (such as GDP increase/decrease) to calculate changes in EAD? If yes, differences in economic growth might be another factor in explaining the observed trends

L347-350: this introduction is perhaps not necessary here.

L367: do you mean 75% higher compared to status-quo insurance?

L378-380: I would tone down this conclusion, perhaps saying " traditional climate risk assessment methods (...), might inaccurately project changes in future flood risk". You indeed showed the importance of endogenous factors in shaping flood exposure and vulnerability, yet several other factors influence flood risk too (structural flood control measures, centralized land-use planning etc).

L418: can you cite here some of these studies?

L625: could you provide a reference for the choice of this discount rate value?

L669-670: The paper by Aerts is a literature review of previous applications of a range of risk reduction measures. Perhaps the authors should provide more details on how they used such information to model costs and effectiveness of flood-proofing measures

Francesco Dottori
CIMA Research Foundation

References

Di Baldassarre, G., et al: An interdisciplinary research agenda to explore the unintended consequences of structural flood protection, *Hydrol. Earth Syst. Sci.*, 22, 5629–5637, 2018.

Dottori, F., Mentaschi, L., Bianchi, A., Alfieri, L., & Feyen, L. (2023). Cost-effective adaptation strategies to rising river flood risk in Europe. *Nature Climate Change*, 1-7.

Haer, T., Husby, T. G., Botzen, W. W. & Aerts, J. C. The safe development paradox: An agent-based model for flood risk under climate change in the European Union. *Global Environmental Change* 60, 102009 (2020).

Tellman, B. et al. Satellite imaging reveals increased proportion of population exposed to floods. *Nature* 596 (7870), 80–86 (2021).

Tiggeloven, T. et al. Global-scale benefit–cost analysis of coastal flood adaptation to different flood risk drivers using structural measures. *Natural Hazards and Earth System Sciences* 20 (4), 1025–1044 (2020).

Vousdoukas, M. I., Mentaschi, L., Hinkel, J., Ward, P. J., Mongelli, I., Ciscar, J. C., & Feyen, L. (2020). Economic motivation for raising coastal flood defenses in Europe. *Nature communications*, 11(1), 2119.

Ward, P. J. et al. A global framework for future costs and benefits of river-flood protection in urban areas. *Nature Climate Change* 7 (9), 642–646 (2017).

Response to reviewer 1:

We greatly value the time and effort the reviewer has put into examining our manuscript. We are particularly grateful for the reviewer's laborious inspection of the methods section, including the model description. We believe that addressing the issues put forward by the reviewer has considerably improved the article.

Based on the reviewer's notes, we thoroughly revised large parts of the methods section, but also addressed several issues in the main text. In the following, we summarize how we addressed the most important issues put forward by the reviewer.

- The description of the decision model (Section A.1.2), in which households' settlement location choices are simulated, is changed almost entirely, and now includes more comprehensive and intuitive equations. Unclear symbology in these equations is addressed, and variables are more precisely explained in text.
- The method to estimate flood risk (Section A.2) is revised completely, as requested by the reviewer. We applied a more systematic approach to describe which variables are estimated here, for which purpose, and how these are derived using a modeling approach that is taken from previous studies.
- How household-level adaptation is integrated in the model is described more elaborately in Section A.1.2. Also, in the main text we discuss more explicitly how household-level DRR may motivate some households to settle in floodplains, and why it is important to consider this in our study on floodplain exposure development. However, this element is not an essential focal point of this study, which is why we do not specifically address the extent to which household-level DRR motivates households to settle in floodplains. Instead, we include it as an option that households have at all times to reduce flood risk to their property.
- The model calibration is discussed more elaborately (Appendix B). In particular, it is discussed explicitly how our model produces results that are approximately aligned with observations in Tellman et al. (2021). Limitations of the calibration process are also discussed in more detail.
- Results presented in the sensitivity analysis (Appendix B) are more critically scrutinized. We agree that in the initial version of the manuscript the discussion of these results was insufficient. In the revised manuscript this is improved.

Besides these general remarks, specific responses to the comments are given below, where the line number refers to the location of the comment in the file provided by the reviewer. We again thank the reviewer for their time and effort, and hope they will enjoy reading the improved manuscript.

Line 50: "*Which processes? not clear*"

We adjusted the text to clarify this. It now states "A more accurate assessment of developments in flood exposure requires a coupling of human and environmental subsystems " (lines 51-52)

Line 90: "*and flood risk*"

Flood risk is considered a disamenity. We clarified this by explicitly mentioning flood risk in brackets. (line 106)

Line 93: "*I think the meaning of SSP2 in terms of population growth should be briefly explained here as in the appendix*"

We revised this paragraph extensively to clarify the interpretation of the results and the use of SSP-scenarios. This is what it looks like now: “To clearly present the flood exposure projections developed in this study, we consistently compare our results to the data obtained from the 2UP-model (Van Huijstee et al., 2018). The 2UP-model applies a spatial suitability map to project population growth on a local scale using generic population growth scenarios, such as SSPs. The method developed in this study differs from the 2UP-model by considering environmental (dis)amenities (including flood risk) and insurance. The population projection using the 2UP-model is henceforth referred to as the “baseline” approach. Figure 1, Panel A, shows the projected population growth within floodplains for the period 2010-2050 using the baseline approach. This projection applies population growth according to SSP2, which represents a continuation of historical trends in terms of social, economic, and technological development.” (lines 101-112)

Line 99: *“What do authors mean with adaptation options? They were not described before and the reader can understand its meaning only reading the appendix”*

We agree that the description of household level DRR should be covered earlier in the manuscript. We adjusted the final paragraph of the introduction to highlight that household-level DRR is an adaptation option that households consider when settling in a floodplain. Also, this sentence is adjusted to specify the adaptation options available, which are wet- and dry- flood-proofing.

Line 102: *“This title is not self-explanatory. The same is valid for panel C”*

The title is adjusted to be more self-explanatory and more aligned with titles in Figure 2. The title for Panel B now states: “Difference with Panel A when considering environmental (dis)amenities and household-level DRR”

Line 102: *“The meaning of the colours (i.e. the scale) in panels B and C should be described also in the caption. Likewise, the meaning of grey areas is clear only after reading the appendix. Please explain.”*

The caption is adjusted to clarify these: “(A) Projected population development in floodplains from 2010 to 2050 in percentages using the baseline-method under SSP2. Green shades indicate a decline in floodplain populations, while orange to red shades indicate an increase. (B) the difference with the baseline-projection when considering environmental (dis)amenities of floodplains and household-level adaptation. (C) the difference with the baseline-projection when considering insurance availability in addition to determinants included in Panel B. In Panels B and C, green shades indicate a lower floodplain population compared to the baseline approach, while yellow to red shades indicate a higher population projection. For regions depicted in grey our method is not applicable due to declining populations across these regions (see Section A.1.1)”

Lines 121 & 126: *“Panel B?”*

Indeed. We changed it accordingly

Line 141: *“Here we have also the influence of adaptation measures that is, however, never discussed. In any case, it is not clear to me how this is evaluated in the model (see also my comment in the appendix). Moreover, it is not clear how and if it is possible to analyse the separated effect of flood risk, environmental amenities and adaptation options.”*

We agree that this part of the analysis is somewhat obscure. We adjusted the introduction and the first paragraph of the result to emphasize that DRR is an adaptation option for households who want to move to a floodplain. Moreover, Section A1.2 is thoroughly revised to clarify how this process is modeled.

Line 212: *“This statement sounds strange to me. Is it not the opposite? Please comment”*

The statement is correct, but the writing may be improved to enhance clarity. The idea of risk-based insurance pricing is that premiums for those facing high risk are not subsidized by policyholders facing lower risk, which means that the premiums are more reflective of the actual flood risk. The costs of insurance, therefore, approach the actual costs of flood risk. We changed the sentence in the following way. “In these countries, flood insurance premiums are, to varying degrees, risk-reflective, which means that the costs related to flooding for floodplain residents with insurance coverage approaches the actual projected flood damage.” (lines 239-243)

Line 291: *“See previous comment on colours meaning”*

The figure caption is adjusted to explain the meaning of colours in this figure.

Line 382: *“In fact, the effect of local amenities is not explicit in the model”*

The reviewer is correct. We removed this statement from the sentence. See the following comment for how we changed the sentence.

Line 383: *“What about adaptation options?”*

DRR is, indeed, an option that impacts the decision to settle in a floodplain for some. I included it in this concluding statement. These sentences now state: “Based on these findings we can conclude that traditional climate risk assessment methods, that assume exogenous exposure and vulnerability developments, may inaccurately project changes in future flood risk. A reason is that households' decisions to settle in floodplains endogenously depend on flood risk, coverage of this risk by insurance, and the option to reduce risk through DRR-measures.” (lines 429-434)

Line 434: *“I think the reader is supported if the authors would use always the same terminology (e.g. decision model as referred to in line 429?)”*

This is addressed. We changed the terminology to “household decision model” (line 501)

Line 435: *“Check grammar... unclear”*

The sentence is changed to improve clarity (lines 503-506)

Line 444: *“how coverage is considered is not clear in the remaining of the text”*

Coverage is removed from this sentence because it is not considered in the model. We do include a deductible of 15% of flood damages, but we do not address the impact of different insurance coverage levels on floodplain settlement.

Line 466: *“What about SSP1?”*

SSP1 is included in this sentence in the revised manuscript

Line 519: “*Which are these variables? Not clear*”

Several variables are drawn randomly from distributions or other ranges. For example, the parameters that modify flood probability and impact to integrate a subjective element in the expected utility framework are drawn randomly from normal distributions. Moreover, the two potential settlement locations, expressed as their relative distance from a river, are randomly chosen, one within the floodplain, and one outside it. We thoroughly revised the methods section where these processes are explained. We removed the description here to limit confusion.

Line 532: “*How the time t influences the results is not clear? Please, specify*”

We thank the reviewer for the elaborate review of the model description. We agree that some elements, including the time aspect (t), was not clearly described in the manuscript. To improve this, we thoroughly revised this section.

Line 532: “*I have difficulties with the adopted symbolism. Also because the subscripts used here do not correspond with those in equations A2 and A3. In fact, A , L and s are independent variables as correctly indicated in A2 and A3. Please homogenize.*”

We agree that the independent variables L and A do not belong in this equation. We removed these from the formula and moved the text describing these variables below Equation A2.

Line 540: “*In fact, the variable s is not used in the following. How it influences the model?*”

After revising Equations A2 and A3, we completely changed the approach to describing the subscript (s) in Equation A1. Households choose to settle in a floodplain if their EU associated with one of three available strategies (s) is higher than the EU of the higher ground. These strategies include no action, apply a DRR-measure, or purchase insurance coverage. This explanation is also integrated in the text.

Line 575: “*Again. Problems with symbolism. Are authors saying that $W\pi$ is equal to $W-\pi$?*”

The reviewer is correct. We adjusted the notation in the manuscript.

Line 576: “*Why L is not reduced if an insurance coverage exist?*”

A good point. We adjusted the formula in the manuscript to show this.

Line 581: “*Alfa?*”

Alfa is the insurance coverage. In the revised manuscript this flood impact misperception parameter is gamma, which is also explained more thoroughly in the text.

Line 590: “*Which values can the parameter take and which is their meaning?*”

We agree that this process is unclearly described. In the revised manuscript, we improved transparency regarding the derivation of the flood impact and probability misperception parameters, and also on which literature these are based.

Line 610: “*What about t ?*”

Indeed, it also impacts expected utility on the high ground. All variables considered change over time. This element is included in Equation A3.

Line 623: *“This step is not clear to me. If amenities values are divided by 15 to consider an annual value, why probability and premiums are multiplied by 15? Please comment”*

We agree that this may be confusing because of the way it is written here. We adjusted this text to clarify this process. “the amenities of river proximity, reflected through housing prices, do not express a yearly value, but rather the benefits of a location over the time a household expects to reside in a property. Therefore, within the SEU-framework it is incorrect to compare a long-term benefit with annual flood or insurance costs. For this reason, the insurance premium in Equation A2 is an aggregation of the yearly premium over the expected time of residence, which is assumed to be 15 years. A residence time of 15 years also means that a flood is more likely to occur compared to a single year. For this reason, the perceived annual flood probability is multiplied by 15 in Equation A2.” (lines 701-709)

Line 626: *“This section is totally unclear. How damage is evaluated is murky. Why a Monte Carlo Analysis is required? I understand this part is not novel but the reader must be able to understand and replicate the methodology. I suggest a deep revision.”*

A deep revision of this section was done. The description of the Monte Carlo simulation is left out because this process is done to approach the integral value of the flood probability-impact curve, which is a practical detail in the modeling approach. The reasoning of why an integral is needed is explained in more detail in the improved manuscript.

Line 663: *“This part is totally not clear. How adaptation options influence expected utility is not described in section A.1.2. Still results always refer to the effect of adaptation options. This influences the comprehensibility of the results by readers.”*

How adaptation (DRR-) measures are integrated in the model is explained in more detail and more intuitively using the expected utility framework in section A1.2. More specifically, applying a DRR-measure is now shown as one of three options that a household may take if settling in a floodplain. A household may be enticed to settle in a floodplain if it is able to limit potential flood damages through DRR. The expected utility of settling in a floodplain and applying a DRR-measure depends on the risk-reduction potential and the costs of the DRR-measure. This is all explained in more detail in Section A1.2

Line 673: *“If adaptation options are implemented W should reduce. Is it correct? Is this taken into account?”*

See comment above and Section A1.2

Line 691: *“Italy does not have an insurance system”*

Insurance against climate-related events exists in Italy, but uptake by households is extremely low (<5%), which may reflect a certain undervaluation of risk, or trust in alternative financing of damages such as government compensation. In previous studies, the DIFI-model was used to project insurance uptake by households considering future flood risk (Tesselaar et al., 2020; Tesselaar et al., 2022), both of which project insurance uptake values that approach reality. Also, in the current study it can be seen that the introduction of insurance dynamics does not change population growth much, which indicates that insurance is not considered an attractive product to limit financial vulnerability to flood risk.

Line 731: *“This part is interesting but not fundamental for the understanding of the method. Considering that the method is complex, I would avoid adding additional information that can create confusion in the reader. Please, consider removing.”*

A good suggestion to remove non-essential information. This paragraph is removed in the manuscript.

Line 737: *“This concept was repeated many times. Consider removing.”*

We agree. These sentences are removed in the revised manuscript.

Line 754: *“What do authors mean with choice? Not clear”*

The “choice” here refers to the decision between the two potential settlement locations. However, we realize that this description skips several important steps to understand the procedure. Since floodplains take different shapes, it is difficult to randomly assign a settlement location for a household within this floodplain, and then determine the distance from the river. At least, this process would be too complex for the purpose at hand. Therefore, we chose to homogenize the shape of all floodplains, to directly surround the river throughout the entire NUTS3-region. The width of the floodplain becomes the average over all floodplains within the region. We realize that “choice” is an inaccurate word to convey this. We changed it to “the surface area of the NUTS3-region”. We also thoroughly revised this paragraph. So, we invite the reviewer to read it and check whether it is explained more clearly in the manuscript. (lines 822-844)

Line 764: *“I am not sure I totally got it. This part should be better explained.”*

We appreciate the reviewer’s request for more clarity. To improve this, we thoroughly revised this paragraph. (lines 822-844)

Line 805: *“check grammar”*

We adjusted the sentence: “Roughly half of all observations show positive and statistically significant effects of natural amenities on house prices.” (881-882)

Line 832: *“Which is the function at the end? in mathematical terms?”*

This section is thoroughly revised and now provides details on the derivation of the amenity value in an equation.

Line 849: *“Why damage functions must be modified is not clear. This is because how damage is calculated is not clear. See my previous comment.”*

We thoroughly revised the description of the flood risk model. We also adjusted the text here to enhance clarity (lines 935-946)

Line 879: *“which are the results of calibration? please briefly comment”*

We added a comparison of our results with those of Tellman et al. (2021), which serves as a calibration method for our simulation. We added the following to the paragraph (lines 969-984):

“Because Tellman et al. (2021) does not show results for some European countries, we performed the calibration on countries that did show results, including France, the UK, Germany, the Czech Republic, Poland, and Romania. Status-quo insurance arrangements were used to compare model output to results in Tellman et al. (2021), meaning flat-rate insurance premiums for France and Romania, and risk-based premiums for all remaining countries. The results of calibration show a slight overestimation from results in Tellman et al. (2021) of, on average over the considered countries, 18%. This overestimation is largely driven by our projection for the UK, which shows a 50% overestimation when considering the current insurance system. However, the flood insurance system in the UK was not yet firmly established in the period 2000-2015 (Surminski et al., 2015), which may explain at least part of the overestimation. Our projection for the UK without insurance availability shows only a slight overestimation of 10%.”

Line 924: *“They look to me different. Please, comment”*

What we mean here is that, although the absolute growth of floodplain populations are higher under SSP5, the overall patterns remain roughly similar. Regions where populations decline in SSP2, also decline in SSP5, and vice versa. It is mainly the extent of projected changes that are larger under SSP5 (i.e., colors become darker green in Panel H and darker red in Panel I). There are very few regions where outcomes change from declining to increasing projections (i.e., from green to red or the other way around when comparing Panels B, E, and H; or Panels C, F, and I). A reason for “stronger” findings under SSP5 is given by the following text that is added to the manuscript: “Whereas deviations in population growth when considering flood risk, adaptation options, and amenities (Panel H) shows a similar pattern to regions in other scenarios (Panels B and E), the extent of these deviations is larger under the RCP8.5-SSP5 scenario (i.e. dark green and bright red colors occur more frequently in Panel H than in panels B and E). One reason for this projection is that overall population growth in European regions is considerably higher under SSP5, as can be seen in Panel G, which means more households face the settlement location choice in our simulation, raising the potential for deviation with the baseline. The main driver of floodplain population growth remains flood risk, which is also more severe under the RCP8.5 scenario in many parts of Europe” (lines 1040-1050)

Line 929: *“I do not agree. Could authors better explain?”*

See the response to the previous comment and the revised paragraph in the manuscript, which goes as follows: “Introducing status-quo insurance coverage (Panel I) means that the negative impact of flood risk on the expected utility of floodplain settlement is considerably reduced, particularly when insurance coverage is cheap (i.e. France and Spain), causing more households to opt for the floodplain amenities. Again, a higher number of households facing the settlement location choice under SSP5 allows for more outstanding results compared to the previous scenarios.” (1050-1056)

Lines 929, 268, & 977: Plot colours and titles are adjusted as requested.

Response to reviewer 2:

First of all, we sincerely thank the reviewer for their time and effort spent on examining the manuscript. We are grateful for the critical perspective of the reviewer regarding the assessment of flood risk, and the presentation and discussion of results. We believe that addressing the reviewer’s critique and questions, amongst other improvements, enabled a more intuitive interpretation of the study’s results. We also thank the reviewer for suggestions made to place the study’s findings in a broader policy-context, which was followed, and clarifies the position of this study within the field of flood risk

management. In the following, we address each of the reviewer's comments point-by-point, where the reviewer's comments are included in italics.

"I find the maps shown in NUTS3-level misleading. The study focuses on river flooding but the spatial aggregation covers the coastal areas as well. The latter has very different flood exposure due to coastal flooding or combined river/coastal flooding. It is not clear if the population exposure obtained by using the 2UP model is also for river flooding and if the population data used do not include population in coastal areas. Fig 1 shows the key results within the main manuscript, but it is too small to see the spatial differences. It needs to be enlarged and ideally the coastal areas should be excluded. The discussion in the manuscript always refers to countries, and as such the country boundaries should be visible on the map."

We appreciate your request for further clarification regarding the differences between coastal flood risk and riverine flood risk, particularly in delta regions. To address this point, it is important to emphasize that our study specifically focuses on riverine flooding and does not incorporate the complexities associated with coastal flooding or combined river/coastal flooding. Our analysis solely considers the excess surface water resulting from riverine flooding, which is modeled using a hydrological model. As a result, we did not take into account any flooding from the coastal side. Furthermore, it is important to note that while our study primarily focuses on riverine flooding, we do account for population in coastal areas that are affected by riverine flooding events. In other words, we consider the population residing in coastal areas that are at risk of flooding due to riverine sources. We added text in the title, abstract, and introduction, to emphasize that the scope of this paper concerns riverine flood risk. We also included a larger version of Figure 1 in Appendix C. Moreover, figures throughout the manuscript are adjusted to show national borders.

"It is not clear where the 100-year floodplains are located. The authors referred to it throughout the manuscript, for example line 119-122."

The 100-year floodplains directly surround river systems. Whereas in some areas floodplains may cover extensive areas, such as in delta regions, in many inland regions floodplains cover a narrow area surrounding a river. The exact location of floodplains can be observed in the Aqueduct Flood Analyzer (<https://www.wri.org/data/aqueduct-global-flood-analyzer>), which applies the same flood hazard model as our study. Because the 100-year floodplain is often a relatively small area, it would not be visible on a continental-scale map. For this reason, we aggregated the floodplain area to NUTS3-level. All figures, except A1-A4, show population growth or flood risk specifically for the 100-year floodplains located in a NUTS3-region. We adjusted the text in lines 116-118 to emphasize that shown results in the manuscript represent all floodplain areas within the NUTS3-region.

"One can hardly see any floodplains in the maps but rather entire countries or regions without much difference in the colours."

See our response to the comment above.

"It was not explained in the manuscript why the 100-year return period was used for the analysis. Surely the more frequent floods such as 50-year would be of more interest unless the authors assume the protection standards are higher than 50-year all over Europe. The second issue is the 100-year flood extrapolated using shorter time slices is not robust."

We agree that the choice to focus on exposure growth specifically in 100-year floodplains needs to be substantiated. In academic literature, but also for policy-design, the 100-year flood is often used to

demarcate the area "at risk" of flooding. For example, the national flood insurance program in the US, as well as the environmental agency in England and Wales apply this standard to direct adaptation policy. We integrated this discussion in lines 114-116, including relevant references. The choice to focus our analysis on 100-year floodplains does not mean that floods of different probabilities cannot occur there. For example, a 50-year flood may occur, although causing less damage, and a 1000-year flood may occur, causing more severe damage. We thoroughly revised Section A.2 to clarify how flood risk within 100-year floodplains is estimated.

“Throughout the paper, the term of ‘risk ’and ‘hazard ’are used interchangeably without making distinction. The authors need to distinguish between ‘flood risk ’and ‘flood hazard ’and define the terms of hazard, vulnerability and exposure that make up ‘flood risk’. The definitions in line 277-280 are not accurate. Hazard is not “inundation of land due to the over-topping of dykes or embankment.” Vulnerability is not just “the extent that exposed assets are damaged by a flood of a certain magnitude.”

The definitions of flood hazard, exposure, and vulnerability that we maintain throughout the article are aligned with definitions used in relevant literature (e.g., Kron, 2009; <https://doi.org/10.1080/02508060508691837>). We do agree with the reviewer that different definitions of these elements exist. For example, besides only focusing on assets, vulnerability and exposure sometimes also considers how floods disrupt economic activity, livelihoods, or ecosystems. However, as this study concerns an application of insurance to physical damages, we define flood risk in this study as damage to the built environment. The thoroughly revised Section A.2 now more intuitively describes the process of the assessment of flood risk in terms of hazard, exposure, and vulnerability. In particular, flood hazard is a river’s water-level associated with a certain return period, which is compared to regional protection standards to assess whether a certain water-level can cause inundation of land; exposure is the built (residential) environment at risk of inundation; and vulnerability is the extent to which the built environment is damages by a certain level of inundation, which is estimated using depth-damage curves.

“One important aspect missing in this study is government intervention in flood risk management. There are flood zones that are prohibited from development or need planning permission. For example, the Environment Agency in England provides a flood zone map and they have specific regulations on what types of building and what kind of planning permission would be required in a certain flood zone. This study seems to assume the population settlement is completely free from government regulations and planning policies. The equation A1 in Section A.1.2 is rather simplistic assuming settlement decision is completely made by individuals without government regulations. Insurance premium could put constraints on floodplain settlement, but simply increasing the premium is not a solution as it would disadvantage certain households and individuals. The message would be misleading if we simply increase insurance premiums with the aim of reducing exposure. I would like to see discussion on the impacts or effects of government intervention in addition to completely insurance policy driven analysis.”

We thank the reviewer for this suggestion to enrich the discussion of the topic in a broader policy context. Indeed, the decision to settle or not in flood-prone areas is influenced by a broader range of factors than just flood risk, amenities, and flood insurance. As suggested, governments may prohibit urban development in flood-prone regions, enforce flood-prone building-standards to improve resilience against floods, or altogether reduce flood risk by improving flood protection infrastructure. We added text in lines 77-82 and 472-483 to place the proposed adaptation of insurance systems within the broader context of flood adaptation. Concerning the impact of risk-based premiums on individual households, a brief reflection and policy solution is given in lines 465-471.

“Further to my comment 1, Figure A4 has major flaws with regards to the floodplains. If the study focuses on river flooding only, it is incorrect to see 100% percentage of floodplain areas along the coasts.

For example, the Rhine, Po river deltas, the Broadlands in east England etc. These areas are dominated by coastal flooding rather than river flooding.”

- 1) While our study primarily focuses on river flooding, it is important to note that we acknowledge the presence of coastal flooding in certain areas. We understand that delta regions, such as the Rhine and Po river deltas, as well as the Broadlands in eastern England, are susceptible to both riverine and coastal flooding. However, due to the limitations of available data and the scope of our study, our analysis solely considers riverine flooding. This approach allows us to provide a comprehensive assessment of flood risk within the context of our research objectives.
- 2) Our study employs a robust hydrological model to simulate riverine flooding based on excess surface water. This model accounts for factors such as rainfall, terrain, and land use to estimate flood risk from riverine sources. By focusing on riverine flooding, we aim to provide a detailed analysis of the specific mechanisms and impacts associated with this type of flooding. Understanding riverine flooding is crucial for effective flood management and mitigation strategies, especially in regions where it is the dominant form of flooding.
- 3) We apologize for any confusion caused by the representation of floodplain areas in Figure A4. It is important to clarify that the floodplain depiction in the figure serves as a visual representation of areas potentially susceptible to riverine flooding. It does not imply that these areas are exclusively affected by riverine flooding. In the text supporting Figure A4, we added a sentence to describe more explicitly how floodplains are determined and that this specifically concerns riverine flooding (lines 825-826).
- 4) As with any research, there are inherent limitations and scope considerations. While our study focuses on river flooding and its associated population exposure, we acknowledge that coastal flooding is an important aspect of overall flood risk in some regions. We believe that future research should explore the complexities of coastal flooding and its interaction with riverine flooding to provide a more comprehensive understanding of flood risk in these specific areas.
- 5) The Rhine river basin, particularly in the Netherlands, is highly susceptible to riverine flood risk. The geography of the Netherlands, with large parts of the country lying below sea level, makes it particularly vulnerable to riverine flooding from rivers such as the Rhine. The Dutch have implemented extensive flood protection measures, including an intricate system of dikes, levees, and flood control structures, to mitigate the risks associated with riverine flooding. However, despite these measures, a significant portion of the Rhine river basin in the Netherlands remains at risk of riverine flooding, with estimates suggesting that over 80% of the region is susceptible to such flood events by our model.

“Fig A1: Flood risk appears in “Impact variables for choice module”, but also in “Model output”. It is very confusing. I think the Flood risk in “Impact variables for choice module” should actually be “Flood hazard” or probabilities, and the Flood risk in “Model output” should actually be “Flood damage” or EAD.”

We agree that mentioning "flood risk" twice in the chart may be confusing. However, both occurrences are truly flood risk estimations, so we cannot follow the reviewer's suggestion to refer to "flood hazard". In particular, flood risk is calculated in "impact variables for choice module" to determine insurance premiums and influence households' decisions whether to settle in a floodplain or not. Using the new estimations of population growth in floodplains, we are interested to observe how flood risk estimations change. For this, we rescale the initial flood damage estimations to the population dynamics per region, and calculate EAD using the same method as before. To emphasize that the estimation of flood risk in "model output" is different from the initial estimation, we adjusted the flowchart to mention "model output: adjusted projections".

“Section A.2 Flood risk model: The heading should change to ‘Flood damage ’or ‘Flood loss ’model because that you describe here is the method obtaining the flood damage and losses. The study uses 100-year return floodplain to estimate population exposure, but the method describe here uses a range of return periods from 2- to up to 1000-year. This is rather confusing. Is the annual damaged derived for 100-year return period only or all the return periods beyond 100-year?”

We understand the confusion about how the flood return-periods are used to estimate flood risk. To clarify this, we thoroughly revised section A.2. We do not see the added value of changing the title of the section to "flood damage" or "flood loss" model. In particular, the approach to calculate EAD considers probabilities (return-periods), which means that "damage" or "loss" would be an incomplete title for the section.

“The manuscript needs English proof reading to make it easier to read. There are sentences that are formulated in strange ways and difficult to understand. Some sentences are too long and can be broken down to shorter sentences.”

A professional English editing service was performed.

“Line 74: the most effective method to limit societal exposure. Suggest change to “the most effective method to limit population exposure” as it is better to stick to your terminology.”

Thank you for the suggestion. We adjusted the text accordingly.

“Line 167-174: Instead of using ‘flood risk ’here, it should be ‘flood losses ’or ‘flood damage’. It is also not clear if this is annual flood damage of 100-year return flood or flood damage of any unprotected magnitude.”

The confusion regarding terminology is hopefully clarified in responses to earlier comments. In this particular case, "flood risk" is the correct term because we discuss flood impacts in annual terms (EAD), which means we consider both damages and probabilities. Mentioning "Flood damage" or "losses" would, therefore, be incorrect. Regarding the second point, we refer to annual flood risk here, which considers flood events of all included probabilities. This process is described intuitively in the revised Section A.2.

“Line 216, This person more likely prefers a lower - Change to ‘Such people more likely prefer’”

Thank you for the suggestion. We adjusted the text accordingly.

“Line 289: fully avoid potential flood damage by moving out of harm’s way - What do you mean by “harm’s way”. Please consider rephrasing here.”

We changed the phrasing here to: ..."may choose to fully avoid potential flood damage by settling on higher ground".

“Line 334: the growth in flood risk - Should be ‘flood damage’”

These numbers reflect the growth in EAD, which is better captured by "flood risk" than "flood damage".

“Line 452: remove ‘, then,’”

Thank you for the suggestion. We adjusted the text accordingly.

“Line 571: flood risk (L); This should be flood losses (L). In line 579, “flood damage (L)” is used. Please be consistent with your terminology.”

This section is revised thoroughly and now consistently uses "flood losses" to describe the variable L.

“Line 618: 15 years; Please explain why using 15 years in your model.”

This choice is based on the average time a household resides in a home before relocating in the UK. In some countries it may be slightly more (e.g. in the Netherlands it is 20 years), or less (e.g. in the US it is 13 years). For our analysis using a single length of time for this purpose reduces complexity and improves our ability to interpret results. We added a discussion of this in the manuscript (line 707), including a reference.

“Line 839: future flood risk; Should change to ‘future flood damage’”

We estimate future EAD, which is better captured by "flood risk" than by "flood damage".

“Line 856: regular procedure is followed; What is the regular procedure?”

This is unclear indeed. We adjusted the text, which now states "After this, the procedure described in Section A.2 is followed to obtain EAD under the different insurance and adaptation scenarios."

Response to reviewer 3:

Firstly, we would like to take this opportunity to thank the reviewer for their effort examining the manuscript. We greatly value the reviewer's critical perspective on the interpretation of results, and suggestions made to improve the discussion of the study's findings. By addressing the reviewer's comments, we believe the manuscript is improved considerably. In the following, we describe how each of the reviewer's comments is addressed in the revised manuscript. The reviewer's comments are recited in italics, followed by our response.

“The actual degree of novelty of the modelling framework and of the outcomes is not completely clear to me. The study builds on a number of already-existing models and studies (GLOFRIS for flood risk analysis, 2UP for general population growth, DIFI for insurance modelling) by adding new components for population growth and flood-proofing of buildings. It is not clear to me whether some results, such as EAD estimations, are taken from previous works from the same Authors or are new, and how much of the previous studies by the Authors overlap with the present work. I would invite the authors to briefly describe their past works in the topic, as well as other relevant literature works (see also the following point), and then specify what are the innovations brought by their new analysis.”

We appreciate the reviewer's request for more information regarding the novelty of the method. Text is added in the introduction to more explicitly describe the difference with previous studies and the added value of this study (lines 90-97). In particular, it is emphasized that data from GLOFRIS is used to estimate flood risk and the size of floodplains. An existing framework for estimating insurance premiums and further insurance market performance, such as insurance demand and moral hazard, is used (Hudson et al., 2019). It is explained that the main novelty of this study is the assessment of flood

exposure development in the context of local flood risk, amenities, and insurance incentives. The 2UP-model is used to provide population growth estimates for the baseline scenario. The current study does not build forth on this model. It is explained in lines 103-106 that the current analysis is designed to provide a more detailed interpretation of population suitability maps as in the 2UP-model and other similar frameworks, which is done by considering local flood risk, river amenities, and the compensation of flood damages through insurance.

“The modelling framework focuses on the role of individual or household-based choices in shaping flood exposure and vulnerability, which is perfectly fine. However, the overall discussion of results seem to overstate the relevance of Authors' findings while overlooking the role of flood risk control measures driven by government policies, such as structural measures, land-use planning and so on. Historically, structural flood control measures (river dikes, dams, detention areas etc) have been determinant in shaping floodplain development (Haer et al., 2016; Di Baldassarre et al., 2018), whereas government policies can strongly influence flood exposure by limiting or excluding further developments in flood-prone areas, so individual decision are only a part of this framework. In addition, a number of recent studies evaluated future risk scenarios resulting from a top-down implementation of these and other measures (Ward et al., 2017, Vousdoukas et al., 2020; Tiggeloven et al., 2020, Dottori et al., 2023, with apologies for the self-citation). My impression is that the role of these measures in shaping flood risk can be, at least, as important as the measures analysed in the present manuscript. Therefore, I would recommend including all these aspects in the introduction and in the discussion of results, in order to put the authors' findings in the right perspective.”

We thank the reviewer for this suggestion to enrich the discussion of the topic in a broader policy context. We also appreciate the information and references given to help us form the discussion. We added several sentences in the introduction (lines 77-84). In the discussion of results we added a paragraph to compare our findings to those of other adaptation measures found in literature (lines 387-404). We added a paragraph in the conclusion to place the proposed adaptation of insurance systems within the general discussion of flood adaptation (lines 472-483).

“In the main text there is no explanation on why only some EU countries/NUTS3 regions are represented in Panels 1B-1C and 2B-2C. My understanding from Section A1.1.1 (L500-501) is that population growth in floodplains is evaluated only in NUTS3 regions with overall increasing growing population trends, but this seems in contrast with values in Figure A2-A (where Italy seems to have positive trends, for instance) and with the projected increases in few areas in eastern countries (e.g. Bucharest, Warsaw, Athens). Could you please explain this point?”

The interpretation of the reviewer is correct. There was a minor coding-mistake, where the overall population growth until 2080 was used instead of 2050 to determine in which regions floodplain populations may grow. This meant that some regions in Italy, amongst others, were unjustly excluded from the analysis, since populations there are projected to grow until 2050, but decline afterwards. The mistake has been corrected.

“I understand that a formal validation of the model is probably not possible. Still, I would ask the Authors to describe how their results compare with the study by Tellman et al after the calibration (in particular, if the model was able to reproduce observed dynamics)”

A formal validation of results is, indeed, not possible. We did, however, revise the text in Appendix B to include a more explicit comparison of our results to those of Tellman et al. (2021) in Appendix B (page 50-51). To do this comparison, an adjusted version of our modeling framework was developed and executed with relevant flood risk and population data for the period 2000-2015. As Tellman et al. (2021)

is a global study, and observations for Europe are limited, the amount of data points on which to compare results is limited. We find that the extent of floodplain population growth matches reasonably well for the countries included in the analysis of Tellman et al. However, our results for the UK are considerably higher (approx. 50%), which may be caused by an inaccurate representation of insurance incentives (a reference for this is given in line 979). Apart from the UK, our results show roughly similar patterns as Tellman et al. In particular, countries that show relatively high floodplain population growth in Tellman et al. (i.e. Germany, Romania) also show higher floodplain population growth compared to other countries in our analysis. Unfortunately future exposure growth projections in Tellman et al. cannot be used as a comparison with our results for the purpose of a validation. This is because Tellman et al. apply the population projection method of the 2UP-model, which is what our method is improving upon.

“Title: my impression is that now the title suggests that insurance is the main driver of population growth in floodplains, which is not the case. Perhaps a more nuanced wording would be more appropriate”

We agree with this suggestion. The title is adjusted to "Flood insurance is a driver of population growth in European floodplains".

“You should clearly specify in the title, abstract and introduction that the work is about river flooding and Europe. Similarly, along the text you should mention whether your statements refer to floods in general (either river, coastal, flash or pluvial) or river floods only.”

The title is adjusted to specify the geographical extent of the study, which is Europe. Text is adjusted in the abstract and throughout the introduction to indicate that the study concerns riverine flood risk and the geographical extent is Europe.

“L93 I would briefly mention here the SSP and RCP scenarios that have been considered in the study.”

We expanded on the explanation of SSPs and RCPs here (lines 110-112 & 122-125), and referred to Appendix B for alternative scenario combinations. Specific information on scenario's shown in the appendices may be excessively detailed to include here.

“Figure 1: Legends are not fully clear in this Figure, I suggest using the same definitions as in Figure 2. Also, for panel A, using a colour palette with different colours for negative - positive change (as in panels B-C), would make the map more readable”

Thank you for these suggestions. They are applied in the revised manuscript.

“L236: “Interestingly, some of these are regions where projections without insurance availability...” Could you list some of these regions here?”

We added three regions in different countries as example (lines 266-267). There are, however, more regions where this is the case.

“L240-243: “... population growth projections in floodplains are consistently and considerably higher in France and Belgium, where premiums are relatively inexpensive in high risk areas.” Could you provide an average monetary value for risk-insensitive and risk-based premiums as estimated by the model?”

We thank the reviewer for the suggestion. The following text was added (lines 273-275): "Whereas average risk-based premiums in 2050 in Sweden, Ireland and the UK are close to €400 annually per household, flat-rate premiums in Spain, France and Belgium are approximately €13 per household."

"Section 2.4: After having read section A.5, my understanding is that you considered damage to residential and commercial areas to calculate EAD, is it correct? If yes, please specify this in the main text."

The interpretation of the reviewer is correct. However, the study focuses solely on residential settlement and damages, which is now clarified better in Section A.2. Since the damages are calculated for the residential sector from the beginning, an additional adjustment to single out the residential sector, as was mentioned in Section A.5, would be incorrect. We removed this adjustment from the model and the manuscript.

"Figure 2: In panel A, using a colour palette with different colours for negative - positive change (as in panels B-C-D), would make the map more readable. Also, can you please explain how "factor" is calculated for panels B,C,D?"

The figure is changed accordingly. "Factor" indicates that the shown results are multiplicative (i.e. a value of 2 indicates a doubling of the EAD compared to Panel A). This is identical to the approach in Figure 1. Therefore, to prevent confusion, "factor" is removed from Figure 2.

"L304-309: Did you consider also projections of economic growth (such as GDP increase/decrease) to calculate changes in EAD? If yes, differences in economic growth might be another factor in explaining the observed trends"

Indeed, GDP-growth affects the value of exposed assets in the food risk model. The relative change in GDP is higher in Eastern Europe under the scenarios and period considered. As suggested this is a likely cause for the observed trend. We have added this description in the manuscript (lines 349-354).

"L347-350: this introduction is perhaps not necessary here."

Agreed. This sentence is removed from the manuscript.

"L367: do you mean 75% higher compared to status-quo insurance?"

Yes, the interpretation is correct. However, this result changed when adjusting the model (when removing the double-counting of residential damages). Please find the revised text in lines 424-428, and the figure on which this conclusion is based in Appendix B.

"L378-380: I would tone down this conclusion, perhaps saying " traditional climate risk assessment methods (...), might inaccurately project changes in future flood risk". You indeed showed the importance of endogenous factors in shaping flood exposure and vulnerability, yet several other factors influence flood risk too (structural flood control measures, centralized land-use planning etc)."

We agree to the request for a more nuanced formulation here and revised the text accordingly (lines 429-434).

"L418:can you cite here some of these studies?"

We appreciate the request for references here. Several references were added (line 485).

"L625: could you provide a reference for the choice of this discount rate value?"

After thorough deliberation, we decided to remove the discounting of aggregate premiums over the time a household resides at a specific location. We motivated this choice in a footnote on page 31:

"Theoretically, future premium payments need to be discounted when expressed in current terms. However, considering the small confined period of 15 years, including a discounting procedure does not change the outcome notably, as was tested using the model. Hence, we chose not to apply discounting to prevent making the model unnecessarily complex. Moreover, the short individual planning horizon of 15 years already captures time preferences to a certain degree." Naturally, the figures and data mentioned in the text represent the output of the updated model.

"L669-670: The paper by Aerts is a literature review of previous applications of a range of risk reduction measures. Perhaps the authors should provide more details on how they used such information to model costs and effectiveness of flood-proofing measures"

We agree that the application of risk reduction measures in the model was somewhat vaguely described. To clarify this better, we thoroughly revised Section A.1.2.

REVIEWERS' COMMENTS

Reviewer #1 (Remarks to the Author):

I thank the authors for the big efforts they made in revising the manuscript. My concerns have been addressed and, in my opinion, the paper is now ready for publication and for stimulating interesting reflections both in the scientific and policy domain. My congratulations! I really enjoyed reading!

Reviewer #2 (Remarks to the Author):

The authors have thoroughly responded to my comments and revised the manuscript. I recommend its publication.

Reviewer #3 (Remarks to the Author):

I thank the Authors for addressing in detail my comments as well as the comments by the other Reviewers. The paper looks considerably improved and in my opinion it can be published.

Response to reviewers' comments

Reviewer #1 (Remarks to the Author):

I thanks the authors for the big efforts they made in revising the manuscript. My concerns have been addressed and, in my opinion, the paper is now ready for publication and for stimulating interesting reflections both in the scientific and policy domain. My congratulations! I really enjoyed reading!

We sincerely thank the reviewer again for the elaborate critical and constructive feedback on the manuscript in the previous round of reviews, and for the kind words in this round. We believe your time and effort substantially improved the work.

Reviewer #2 (Remarks to the Author):

The authors have thoroughly responded to my comments and revised the manuscript. I recommend its publication.

We sincerely thank the reviewer again for the elaborate critical and constructive feedback on the manuscript in the previous round of reviews, and for the kind words in this round. We believe your time and effort substantially improved the work.

Reviewer #3 (Remarks to the Author):

I thank to Authors for addressing in detail my comments as well as the comments by the other Reviewers. The paper looks considerably improved and in my opinion it can be published.

We sincerely thank the reviewer again for the elaborate critical and constructive feedback on the manuscript in the previous round of reviews, and for the kind words in this round. We believe your time and effort substantially improved the work.